# A *nonS-locus F-box* gene breaks self-incompatibility in diploid potatoes

Ling Ma[1,7], Chunzhi Zhang[2,7], Bo Zhang[1,7], Fei Tang[1], Futing Li[1], Qinggang Liao[2], Die Tang[2], Zhen Peng[3], Yuxin Jia[2], Meng Gao[1], Han Guo[4], Jinzhe Zhang[5], Xuming Luo[2], Huiqin Yang[1], Dongli Gao[1], William J. Lucas[2,6], Canhui Li[1✉], Sanwen Huang [2✉] & Yi Shang [1✉]

Potato is the third most important staple food crop. To address challenges associated with global food security, a hybrid potato breeding system, aimed at converting potato from a tuber-propagated tetraploid crop into a seed-propagated diploid crop through crossing inbred lines, is under development. However, given that most diploid potatoes are self-incompatible, this represents a major obstacle which needs to be addressed in order to develop inbred lines. Here, we report on a self-compatible diploid potato, RH89-039-16 (RH), which can efficiently induce a mating transition from self-incompatibility to self-compatibility, when crossed to self-incompatible lines. We identify the *S-locusinhibitor* (*Sli*) gene in RH, capable of interacting with multiple allelic variants of the pistil-specific S-ribonucleases (S-RNases). Further, *Sli* gene functions like a general S-RNase inhibitor, to impart SC to RH and other self-incompatible potatoes. Discovery of *Sli* now offers a path forward for the diploid hybrid breeding program.

[1] Key Laboratory for Potato Biology of Yunnan Province, The CAAS-YNNU-YINMORE Joint Academy of Potato Science, Yunnan Normal University, Kunming, China. [2] Shenzhen Branch, Guangdong Laboratory of Lingnan Modern Agriculture, Genome Analysis Laboratory of the Ministry of Agriculture and Rural Affairs, Agricultural Genomics Institute at Shenzhen, Chinese Academy of Agricultural Sciences, Shenzhen, China. [3] College of Plant Science and Technology, Beijing University of Agriculture, Beijing, China. [4] Department of Economic Plants and Biotechnology, Yunnan Key Laboratory for Wild Plant Resources, Kunming Institute of Botany, Chinese Academy of Sciences, Kunming, China. [5] Key Laboratory of Biology and Genetic Improvement of Horticultural Crops of Ministry of Agriculture, Sino-Dutch Joint Lab of Horticultural Genomics, Institute of Vegetables and Flowers, Chinese Academy of Agricultural Sciences, Beijing, China. [6] Department of Plant Biology, College of Biological Sciences, University of California, Davis, CA, USA. [7] These authors contributed equally: Ling Ma, Chunzhi Zhang, Bo Zhang. ✉email: ch2010201@163.com; huangsanwen@caas.cn; shangyi@ynnu.edu.cn

Potato is consumed as a staple food by approximately 1.3 billion people[1]. Currently, most potato cultivars are tetraploids, carrying four copies of chromosome sets, thereby complicating breeding efforts to introduce new traits. This constitutes a major challenge in terms of expanding potato's contribution to feeding the fast-growing global population, especially under the threat of climate change[2].

A potential pathway to circumvent this impediment of potato breeding is the development of a hybrid potato breeding system[3,4], which would convert potato into a diploid seed crop by crossing elite inbred lines. In diploid potato, beneficial alleles can be efficiently introgressed into these inbred lines, thereby shortening the breeding period[3,5]. Moreover, the hybrid offspring, propagated via true seeds rather than tubers, would facilitate both botanical seed storage and delivery, along with eliminating the risks presently associated with tuber-borne pathogens[6]. However, given that a majority of diploid potatoes are self-incompatible, this obligate outcrossing mating system poses a major obstacle (another big hurdle is inbreeding depression, including poor vigor and low fertility in potato) to develop diploid inbred lines[7–10].

In Solanaceae, the self-incompatibility (SI) system is gametophytic and controlled by a single polymorphic locus, called S-locus[11]. This locus encodes two types of determinants: female/pistil S-determinant (a cytotoxic S-ribonuclease, S-RNase)[12] and male/pollen S-determinant (a set of pollen-specific S-locus F-box proteins, SLFs)[13]. The S-RNase inhibits the growth of self-pollen tubes by either degrading ribosomal RNA (rRNA)[14] or disrupting the dynamic equilibrium of the cytoskeleton[15]. Based on the collaborative non-self recognition system, the S-RNase could be polyubiquitinated and degraded by the interactions with the collective action of 16–20 nonself SLFs that induce compatibility[16,17]. Otherwise, the S-RNase would not be degraded, as it is not recognized by the self SLFs[18]. Thus, loss-of-function mutations of S-RNase[8,9], and degradation of this protein by the introduction of nonself SLFs[19], have been reported to confer self-compatibility (SC) in Solanaceae. However, as genetically modified crops are still widely prohibited and even the acceptance of genome-edited crop is under debate, alternative strategies for addressing SI in the development of diploid inbred lines are preferred.

Previous studies have shown that a dominant locus, S-locus inhibitor (Sli), from a wild diploid potato species, Solanum chacoense Bitt., is responsible for breaking down the stylar incompatible response[7,20–22]. Although Eggers et al. reported the identification of Sli from the wild species in the Solanaceae International Online Meeting[23], the molecular mechanism of Sli was not described. Furthermore, this wild accession produces long stolons and high levels of toxic steroidal glycoalkaloids in tubers[8,24,25], which also limits its breeding potential. Alternatively, RH89-039-16 (RH), a long-day adaptive line derived from S. tuberosum group Tuberosum, is another self-compatible diploid potato that produces tubers[26,27]. Clot et al. has shown that RH also harbors the Sli SC haplotype at the distal end of chromosome 12[22]. However, the molecular basis underlying the SI to SC transition in RH remains unknown, and it is unknown whether RH can transmit heritable SC to self-incompatible diploid potatoes.

In this study, we identify a non-S-locus F-box (NSF) gene in RH. This gene is identical to the Sli gene, and capable of interacting with multiple allelic variants of S-RNases, functioning like a general S-locus inhibitor to introduce SC to both RH and other self-incompatible lines. The discovery of Sli offers a new and effective path for the hybrid diploid potato breeding program.

## Results

**SC gene colocalizes with a segregation distortion region.** As shown in Fig. 1a, self-pollinated RH plants produced fruits, and their pollen tubes penetrated the style and reached the ovary. To explore the genetic basis underlying the SC phenotype of RH, F$_1$ hybrids (PI 225689 × RH) from RH crossed with PI 225689, a self-incompatible diploid line (a/a) from S. tuberosum group Phureja, were evaluated for self-compatibility (Fig. 1a and Supplementary Fig. 1a). Genetic analysis of F$_1$ progeny established a ~1:1 segregation ratio of SC to SI (131 self-compatible plants versus 107 self-incompatible plants; $\chi^2 = 2.42 < \chi^2_{0.05} = 3.84$), indicating that SC in RH is caused by a single dominant heterozygous gene (A/a) or gametophytic factor. Next, 40 self-compatible and 40 self-incompatible F$_1$ individuals were carefully selected for a bulked segregant analysis (BSA), which exhibited a peak associated with SC/SI located at the end of chromosome 12 (58~61 Mb, Fig. 1b). InDel markers were then developed, and a total of 238 self-compatible and self-incompatible F$_1$ individuals were used to pinpoint the location of the candidate to an interval between the marker M-1 and M-2 (2.24 Mb, Fig. 1c).

Accurate evaluation of self-compatible/incompatible individuals from mapping populations is essential for the fine mapping of the SC gene. Unfortunately, this phenotyping process is labor-intensive and time-consuming, which may explain why the most established SC gene Sli has not been cloned since it was first reported in 1998[20,21]. Interestingly, we observed an extreme segregation distortion (SD), in that only homozygous haplotype A/A or heterozygote A/a, but no homozygous haplotype a/a, was identified in the F$_2$ progeny from crossing PI 225689 and RH (A/A:A/a = 3030:2954 ≈ 1:1, $\chi^2 = 0.97 < \chi^2_{0.05} = 3.84$). Furthermore, the same SD region was also observed in the S$_1$ population of RH[27,28]. Based on these observations, we hypothesized that selfing of heterozygote self-compatible plants (A/a) would result in SD of the progeny, and the absence of homozygous haplotype a/a in the offspring is linked either to lethal genes or to failed double fertilization resulting from the gametophytic SI in potato. Since RH may harbor SLFs that recognize the S-RNase harbored by PI 225689, individuals carrying the homozygous haplotype a/a were detected in the F$_1$ population (PI 225689 × RH). This result suggests that the SD in the F$_2$ population and the S$_1$ population of RH are caused by the gametophytic SI. Thus, only the pollen harboring the SC gene could penetrate the self-style and fulfil the fertilization to produce progeny, and all the F$_2$ progeny would carry the SC gene and exhibit self-compatible phenotype (Fig. 1d).

**SC in RH is controlled by a non-S-locus F-box gene.** The discovery of the SD in the F$_2$ population greatly accelerated the mapping of the SC gene in RH, by removing the laborious phenotyping requirement. In total, 6624 F$_2$ individuals produced from the 131 selfed F$_1$ plants were used to narrow down the SC gene to a 64.61 kb interval (Fig. 1c). There were five annotated genes in this region (Fig. 1c), and two of them were expressed in pollen (Supplementary Fig. 1b). Interestingly, sequence variations including a 536 bp insertion at 100 bp upstream of the start codon of PGSC0003DMG400016861, encoding an F-box gene, were observed between RH and PI 225689 (Fig. 1e). In addition, quantitative real-time RT-PCR (qRT-PCR) revealed that this gene was highly expressed in RH pollen, but not in the pollen of PI 225689 (Fig. 1f). Based on these findings, PGSC0003DMG400016861 was considered to be the target gene and was named, hereafter, non-S-locus F-box (NSF) gene.

To confirm the in vivo role of NSF, a 3839 bp DNA sequence, including a 2,100 bp promoter region (Fig. 1g), was cloned from RH and transformed into a self-incompatible diploid potato clone from S. phureja S15–65. A population of 3750 explants was screened, from which 26 transgenic positive lines were selected for further study. Flow cytometry assays revealed that only two of the 26 lines

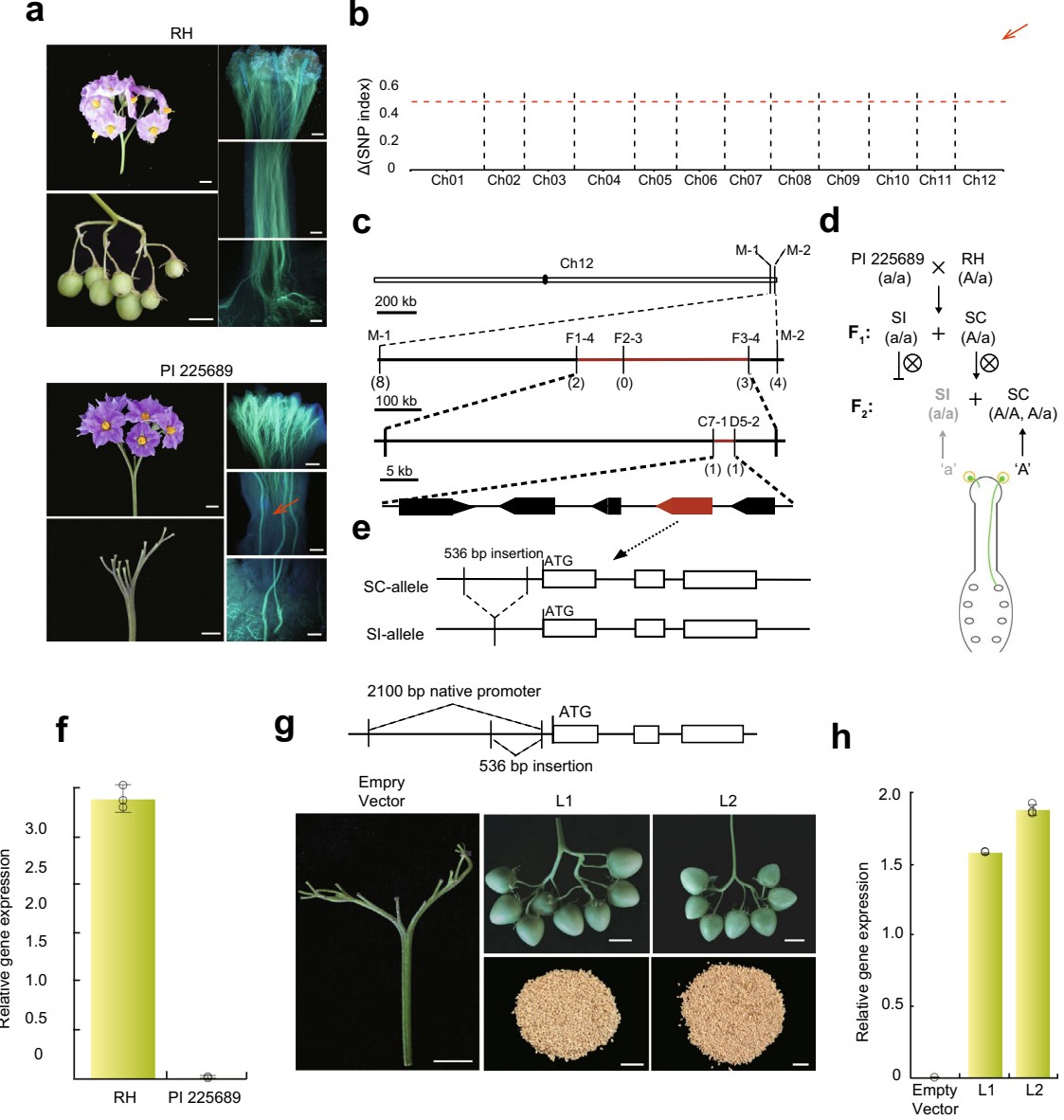

**Fig. 1 Identification of *NSF* by map-based cloning and transgenic assay. a** Developmental status of flower, fruit, and pollen tube, after self-pollination of two parental lines: RH (self-compatibility, SC) and PI 225689 (self-incompatibility, SI). The bar is 1 cm on the four left images and 500 μm on the six right images. The arrested pollen tube is identified with the red arrow. The experiments were repeated five times with similar results. **b** BSA-Seq analysis of the F₁ hybrid population (PI 225689 × RH). The SNP index (Δ) of SC and SI pools, prepared from the F₁ progeny, as mapped along the RH genome. The red arrow indicates a peak above the threshold value. **c** Fine mapping of *NSF* using 238 F₁ and 6,624 F₂ individuals. The number in the bracket indicates the number of recombinants with the same genotype. The *NSF* location is indicated in red. **d** The pollen elongation cease model is proposed to explain the distorted segregation of the F₂ population. Pollen with 'a' genotype is short of SC gene. Cease of pollen tube elongation resulted in unsuccessful fertilization. Thus, the individuals with 'aa' genotype (indicated in pale grey) cannot be identified in the F₂ population. **e** Allelic variation of the candidate gene, *PGSC0003DMG400016861*, between RH (SC) and PI 225689 (SI). The major difference between the two alleles was a 536 bp deletion within the promoter region of the candidate gene. **f** Expression level of the candidate gene, in the pollen of RH and PI 225689, normalized against an endogenous reference gene (*StEF1-α*, elongation factor 1α). Data are presented as means ± s. d. (*n* = 3 biological replicates). **g** Phenotype transition, from SI to SC, after transformation of the candidate gene into a self-incompatible line S15–65. Plants transformed with an empty vector served as a negative control. Note the presence of fruit in L1 and L2 and absence in the vector control. L1/2, two *NSF* transgenic lines. The bar is 1 cm. **h** Expression level of the candidate gene in the pollen of negative control and two transgenic lines, normalized against *StEF1-α*. Data are presented as means ± s. d. (*n* = 3 biological replicates).

were diploid plants; the others were tetraploid (Supplementary Fig. 2). Consistent with our hypothesis, a phenotype change from SI to SC (Fig. 1g), and high *NSF* expression in pollen was observed in these two transgenic lines (Fig. 1h). In contrast, the SI phenotype of the plant transformed with an empty vector was not changed (Fig. 1g), suggesting that high expression of *NSF* in the pollen tissue is required to render SC to RH.

**NSF/Sli is efficient in rendering SC to self-incompatible diploid potatoes.** The most established method of conferring SC is with the dominant *Sli* gene[22], identified in chc 525-3, a self-compatible clone of the self-incompatible species *S. chacoense*[20]. As the *Sli* gene was also mapped at the distal end of chromosome 12 in *S. chacoense*[21,22], *NSF* in RH could be identical to the *Sli* gene in *S. chacoense*. This possibility was tested by analyzing the collinear

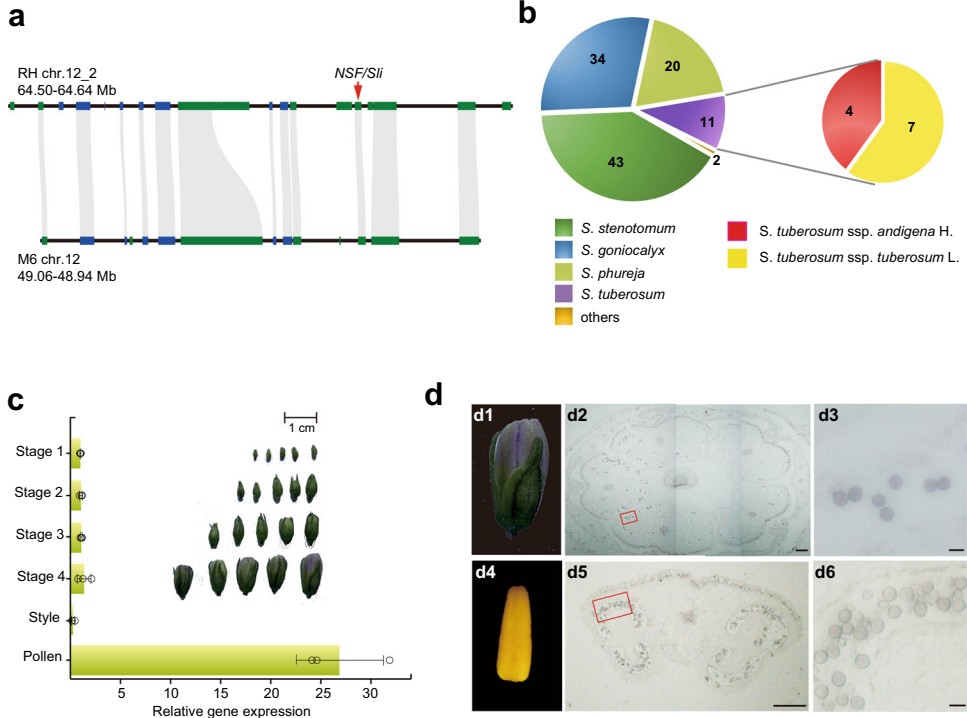

**Fig. 2 Characteristics of *NSF/Sli* in rendering SC to self-incompatible diploid potatoes. a** Collinearity analysis of the distal ends of chromosome 12 between RH and M6. Orthologous relations between the different syntenic loci are depicted by gray. The *NSF* gene is identified with the red arrow. **b** The number of self-compatible diploid potatoes induced by *NSF/Sli*. Six groups (*S. stenotomum, S. goniocalyx, S. tuberosom, S. phureja, S. ajanhuiri,* and *S. curtilobum*) and two subgroups (*tuberosum* and *andigena*) are indicated with different colors. The number represents the accessions that transfer from SI to SC after crossing with RH or E172. **c** Expression patterns of *Sli* in different developmental stages of RH flowers. Development of the RH flower was divided into five stages, including four developing stages according to bud size (status of each stage was shown as illustration) and one mature stage. Expression level was normalized to the expression in stage 1. *StEF1-α* was set as an endogenous reference gene. Data are presented as means ± s. d. (*n* = 3 biological replicates). **d** In situ mRNA hybridization of *Sli* in the stage 4 flower bud and mature stamen. d1 and d4, the stage 4 flower bud (d1) and mature pollen (d4) used in the in situ hybridization assay. d2 and d5, the transverse sections of the stage 4 flower bud (d2) and mature stamen (d5). Bar 200 μm. Magnification of the red square in d2 and d5 are presented in d3 and d6, respectively (bar is 20 μm). The experiments were repeated three times with similar results.

region at the distal end of chromosome 12 between RH and M6, a self-compatible inbred clone created by seven generations of self-pollination of S. chacoense[24,25]. A total of 12 collinear genes, including *NSF/Sli*, and a 120 kb synteny region were identified (Fig. 2a). We further amplified the collinear region, including the 2100 bp promoter and the 1810 bp gene coding sequence, where *NSF/Sli* is located in E172[29] (a self-compatible F$_1$ individual generated by crossing a self-incompatible diploid line E and chc 525-3). The promoter sequences in E172 and M6 are identical to that of RH, but there are five DNA base differences within the *NSF/Sli* coding region, resulting in two amino acid changes (C214R and Q249R) in M6 (Supplementary Fig. 3). Based on this result, we speculated that *NSF* is the *Sli* gene and confers the SC in M6 and E172.

To test whether the *Sli* gene is an efficient pollen SI-breaker, RH and E172 were used as the *Sli* gene donor to pollinate 125 self-incompatible diploid lines that belong to six groups (50 S. stenotomum, 37 S. goniocalyx, 25 S. phureja 11 S. tuberosom, 1 S. ajanhuiri, and 1 S. curtilobum lines, Fig. 2b and Supplementary Table 1). After pollination, 110 F$_1$ populations of the 125 crossings exhibited SC, including 43 S. stenotomum, 34 S. goniocalyx, 20 S. phureja, 11 S. tuberosum (7 tuberosum and 4 andigena), 1 S. ajanhuiri, and 1 S. curtilobum lines (Fig. 2b, Supplementary Table 1). The remaining 15 diploid lines failed to cross with RH or E172 because of different flowering times when compared to that of RH or E172, resulting in unsuccessful pollination. In conclusion, the *Sli* gene in RH

can serve as an efficient SI-breaker to produce self-compatible diploid potatoes.

**Sli is capable of interacting with a diverse set of S-RNases.** Since *Sli* is specifically expressed in floral organs, and its transcript gradually increases with the development of pollen, reaching its maximum expression in the mature pollens' wall (Fig. 2c, d), Sli may play an essential role in mediating the interactions between pollen and stigma, and function like SLFs to interact with and detoxify the S-RNases during pollen tube elongation, to confer SC to self-incompatible lines. To test this hypothesis, firstly, 14 full-length variants of *S-RNases* (StS-RNase1–14), sharing an amino acid identity ranging from 39.62% to 87.38%, were cloned from diploid potatoes using the method described in Ye et al.[8] (Supplementary Table 2). The phylogenetic analysis determined that *StS-RNase1–14* and 35 potato *S-RNase* alleles obtained from Genebank belong to the class III type RNase (Fig. 3a and Supplementary Table 3). Thus, *StS-RNase1–14* were considered as the potential female S-determinants, and used in the following assays.

The motifs located downstream of the F-box domain can confer substrate specificity for ubiquitination[30]. Phylogenetic analysis of Sli revealed a Phloem Protein 2 (PP2) domain with ~160 amino acid residues at its C-terminus (Supplementary Fig. 4). Thus, the C-terminus containing the PP2 domain or the full length of Sli was used in yeast two-hybrid (Y2H) assay to assess the interactions between Sli and the S-RNase variants. As revealed by Y2H, the C-terminal region of Sli interacted with the

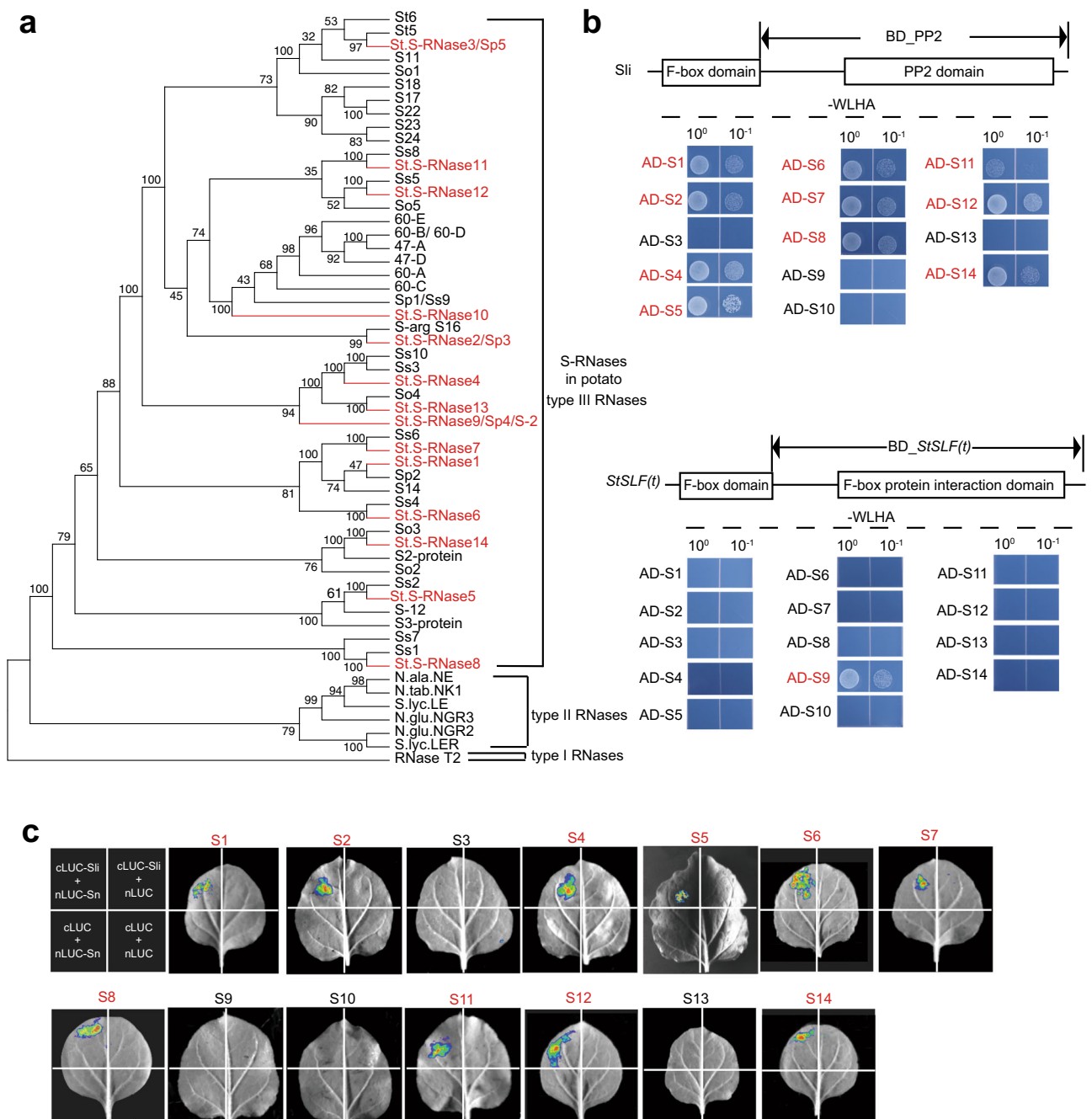

**Fig. 3 Wide interactions between Sli and 14 types of potential StS-RNases. a** A phylogenetic tree analysis of the potatoes S-RNases. StS-RNase1–14 are 14 types of S-RNases reported in this study (indicated in red), while the other 35 S-RNases were obtained from GenBank. **b** Yeast two-hybrid (Y2H) analysis of the interactions between 14 potential S-RNases and the PP2 domain of Sli (BD-PP2). The conserved domains of Sli are indicated by rectangles. The interactions between potential S-RNases and the full length of Sli are shown in Supplementary Fig. 6. The F-box protein interaction domain of a *SLF* gene (*RHC01H2G1617*) was used as a negative control [BD-StSLF(t)]. WLHA, synthetic dropout media lacking tryptophan, leucine, histidine, and adenine. BD binding domain, AD activation domain. **c** The firefly luciferase (LUC) complementation imaging assays of the interactions between the S-RNases and the PP2 domain of Sli. LUC signal was captured in the leaf area co-expressing the PP2 domain of Sli and each of the S-RNases (indicated in red), except for S3, S9, S10, and S13. Sn denote each type of the StS-RNase1–14, nLUC the N terminal of LUC protein, cLUC the C terminal of LUC protein. The experiments were repeated three times with similar results.

majority of the StS-RNases, with the exception of StS-RNase3, 9, 10, and 13 (Fig. 3b), while full-length Sli failed to interact with StS-RNase3, 9, 11, 13, and 14 (Supplementary Fig. 5). The in vivo interactions between Sli and the StS-RNase1–14 were further tested using a firefly luciferase (LUC) complementation assay. Here, a fluorescent signal was detected in tobacco leaves co-expressing the N-terminal half of Luc (nLUC) plus the PP2

domain of Sli fusion protein, together with the C-terminal half of Luc (cLUC) plus each of the StS-RNase1, 2, 4–8, 11, 12, and 14 fusion protein (Fig. 3c). In contrast, the C-terminal region of a male *SLF* gene, *RHC01H2G1617*, selected from the 32 pollen-specific *F-box* genes in chromosome 1 (Supplementary Fig. 6), could only interact with StS-RNase9 (Fig. 3b). This is consistent with the collaborative nonself-recognition model that one SLF

generally interacts with limited types of nonself S-RNases, and thus plants need to carry multiple types of SLFs in one S-haplotype to ensure the detoxification of nonself S-RNases, as well as out-crossing in Solanaceae[16].

## Discussion

The extensive interactions between various StS-RNases and Sli provide insight into the molecular basis of the phenotype transition from SI to SC in diploid potatoes. Support for such a model is presented by the facts that both Sli and SLFs belong to the F-box family of proteins, and that the collaborative nonself recognition model is conserved in Solanaceae[11]. It is very interesting to note that, in RH and M6, the SC arose from the evolution of a non-S-locus F box protein, not the novel SLFs as suggested in the two-step model[11,31]. Since Sli interacts not only with the self S-RNase in RH, but also with multiple types of the S-RNases to promote out-crossing with other diploids, the appearance of Sli may represent an efficient route to introduce a fixed SC phenotype to S-RNase-based self-incompatible plants (Fig. 4). It is also interesting to note that all the three self-compatible lines (RH, M6, and E172) harbor a 536 bp insertion at the promoter of Sli (Fig. 1e, f). Although several SNP mutations were also identified within this region in self-compatible and self-incompatible lines, the 536 bp fragment is most likely to be involved in activating the expression of Sli in the pollen of self-compatible lines. Testing of this hypothesis would provide insights into the regulatory mechanism, as well as the evolutionary trajectory of the Sli gene.

Although the Sli gene has been widely used in conferring SC, the fact that Sli cannot interact with all types of S-RNases indicates that the Sli gene cannot be used to confer SC to all the self-incompatible lines. On the other hand, loss-of-function mutations of the S-RNase have been shown to produce a stable SC phenotype in S. pennellii, S. habrochaites, and S. arcanum[18]. These non-GMO inactivating mutations thus provide another efficient method to break the SI of diploids. In our breeding program, we observed a semi-self-compatible diploid, PG6359, due to the low expression of $S_{s11}$-RNase in this line[28]. Since Sli is located on Ch12 and S-RNase is on Ch01, usage of SC genes from different sources in breeding can effectively avoid the genetic bottlenecks caused by the deleterious mutations linked to these loci, considering that S-RNase is near the centromere while Sli is near the telomere.

In summary, the identification of RH as an efficient SC inducer, the general S-locus inhibitor Sli, as well as the markers associated with the SC/SI phenotype would pave the way for developing the inbred lines and accelerate the hybrid diploid potato breeding.

## Methods

**Plant materials.** The self-compatible material chc525-3 (Solanum chacoense Bitt.) was kindly provided by Prof. K. Hosaka (Obihiro University of Agriculture and Veterinary Medicine) and we selected E172 from a vigorous self-compatible clone in the selfed progeny of chc525-3 to cross with E[32]. The RH was kindly provided by the Department of Plant Breeding at Wageningen University. The 115 self-incompatible lines and S15–65 were kindly provided by the International Potato Center (CIP). The self-compatible line M6 was kindly provided by the U. S.

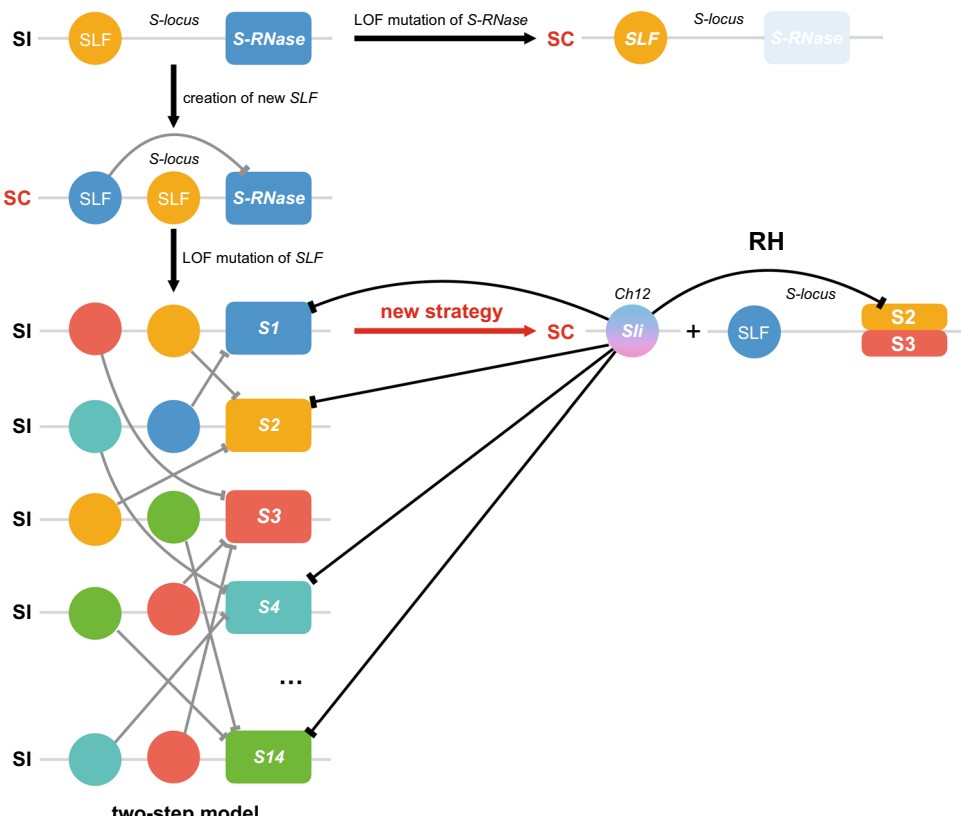

**Fig. 4 A model depicting the evolution of *Sli* in diploid potatoes.** In Solanaceae, a two-step model was proposed to explain the evolution of new SLFs to confer SC to self-incompatible plants, by first interacting with the self S-RNase. Then, the reverse from SC to SI is induced by the loss of ability for these new SLFs to interact with the self S-RNase. The appearance of *Sli* at the non-S-locus region represents a new strategy, in addition to the loss-of-function (LOF) mutations of *S-RNases*, to confer SC to self-incompatible lines. The S-haplotypes in RH are StS-RNases2 and 3. Negative regulations of the multiple StS-RNases by the nonself SLFs or Sli are indicated by the barred lines, which constitute the molecular basis to impart SC to RH and other self-incompatible potatoes. SI self-incompatibility, SC self-compatibility, S1–S14 14 types of S-RNases identified from self-incompatible diploid potatoes, Ch chromosome.

Department of Agriculture (USDA). All of the diploid lines were grown normally in a controlled glasshouse with a 16 h photoperiod and 25 °C day/18 °C night temperatures. Self-incompatible PI 225689 was crossed with RH to produce the self-compatible $F_1$ individuals that were later selfed to obtain the 6624 $F_2$ population for the map-based cloning of *Sli*.

**SC/SI phenotyping**. Mature pollen of the $F_1$ hybrids from RH crossed with PI 225689 (PI 225689×RH) or RH/E172 crossed with the 125 self-incompatible lines was freshly collected and used to pollinate stigmas of the same plant at the blooming stage. Three clones were used to test the SC/SI phenotype for each genotype. This pollination process was repeated at least 30 times for each clone. If all the three clones failed to set berries, this genotype was termed as SI type, while if all the three clones had more than two berries, they were regarded as SC type. Whenever one of the three clones had only one berry, the genotype was regarded as U (uncertain) type.

**Bulked segregant analysis sequencing (BSA-Seq)**. PI 225689 was crossed with RH to produce the $F_1$ individuals that exhibited a 1:1 segregation ratio of SC to SI types. Four DNA pools were prepared using the female parent (PI 225689), the male parent (RH), 40 self-incompatible $F_1$ individuals, and 40 self-compatible $F_1$ plants. The sequencing of four libraries was performed using NovaSeq 6000 platform (Anoroad Gene Technology, Beijing, China). Raw reads were supplied by Anoroad, and clean reads were obtained by quality evolution. Then, the filtered short reads were aligned against the potato reference DM genome (V4.03), the variant calling of SNPs, and small Indels were excavated by Platypus (V0.7.4) software. The heterozygous and inconsistent SNPs, and small indels between the two $F_1$ pools were selected for calculating the Δ (SNP-index) values.

**Fine mapping of *Sli***. To fine map the *Sli* locus, recombinants for the Indel marker M-1 and M-2 were screened using 238 plants of $F_1$ population (created by a cross between PI 225689 and RH). Considering the SC/SI type of the recombinants of $F_1$ plants, the *Sli* locus was further mapped to the interval between F1–4 and F3–4 with a physical interval of 941 Kb. Genotyping of 1800 $F_2$ progenies from 23 selfed $F_1$ plants revealed an extreme segregation distortion (no a/a haplotype). Thus, 6624 $F_2$ individuals from 131 selfed $F_1$ plants were screened for the recombinants, and the *Sli* locus was mapped to the interval between C7-1 and D5-2 with a physical interval of 64.61 kb. The genomic DNA fragments of the interval in RH and PI 225689 were sequenced and aligned. The polymorphic markers were developed based on the re-sequencing data of RH and PI 225689 according to the method described by Zhang et al.[28]. The markers and primers used in this analysis are listed in Supplementary Table 2.

**Gene transformation of S15–65**. For functional identification of the *Sli* gene, DNA fragments containing the native promoter and genomic sequence of *Sli* were amplified from RH, and then inserted into BamH I and Hind III sites of the binary vector pCAMBIA1301. This construct was transformed into the *Agrobacterium tumefaciens* strain GV3101. The S15–65, a diploid self-incompatible line from *S. tuberosum* group Phureja, was kindly provided by CIP and used in the *Agrobacterium*-mediated transformation of potato internodes as previously described[8]. The ploidy of transgenic lines was analyzed by a flow cytometry assay, and the expression level of *Sli* in the diploid transgenic lines was verified by qRT-PCR. SC/SI phenotypes of the diploid transgenic lines were recorded. The primers used in this analysis are listed in Supplementary Table 2.

**Aniline blue fluorochrome staining of pollen tubes**. Five pistils were collected 72 h post pollination and immediately treated with 3:1 95% ethanol:glacial acetic acid. After rinsing twice with double-distilled water, the pistils were incubated in 0.005 mg/mL aniline blue fluorochrome in 0.1 M $K_2HPO_4$ buffer for 24 h. The colored pistil was placed on a glass slide and viewed under a fluorescence microscope.

**Quantitative RT-PCR analysis**. Total RNA was isolated using a RNAprep Pure plant Kit (Tiangen) according to the manufacturer's protocols. First-strand cDNA was synthesized from 1 μg total RNA using the PrimeScript 1st strand cDNA Synthesis Kit (Takara). PCRs were performed using SYBR Premix (Takara) on a 7500 Fast Real-Time PCR system (Applied Biosystems) according to the manufacturer's instructions. Three independent biological experiments were performed in all cases. *StEF1-α* was used as an internal control gene for normalization. The relative gene expression was calculated using $2^{-\Delta\Delta Ct}$. The primers used in this analysis are listed in Supplementary Table 2.

**In situ hybridization**. The 256-bp *Sli*-specific probe was amplified and labeled using the DIG RNA Labeling Kit according to the manufacturer's recommendation (SP6/T7, Roche). Pretreatment of tissue sections, hybridization, and immunological detection were performed in accordance with the previously described methods[33]. The primer pairs used are listed in Supplementary Table 2.

**Phylogenetic analysis of Sli**. The protein sequence of Sli was queried against Phytozome v13 (https://phytozome-next.jgi.doe.gov/) to retrieve its orthologs or homologs from other dicot and monocot species. Protein sequences showing a similarity of 40% or above and evalue lower than 1e-10 blastp to the query sequence were retrieved and multiple aligned using Mafft (https://mafft.cbrc.jp/alignment/software/). Only the two most similar sequences were retained in each species. For SLFs, the F-box protein located within the S-locus region of RH was considered as the potential SLF (http://solanaceae.plantbiology.msu.edu/rh_potato_download.shtml). A maximum likelihood (ML) phylogenetic tree was constructed using the ML based on the Poisson correction model. The bootstrap consensus tree inferred from 1000 replicates was taken to represent the evolutionary history of *Sli*.

**Identification of *S-RNases* from diploid lines**. Specific primers were designed according to the de novo assembly of RNA-sequencing data from the mature styles of the 125 diploid lines using the method described by Ye et al.[8]. The cDNA sample prepared from the styles of each diploid line was used to clone the pistil-specific S-RNase. The positive PCR products were then sequenced and considered as the multiple allelic variants of S-RNases. The primers used in this analysis are listed in Supplementary Table 2.

**Yeast two-hybrid assay**. The codon region of each *S-RNase* (S1-S14) except for its signal peptide was cloned into the pGADT7 vector (AD-*S-RNase*). The C-terminal region harboring the PP2 domain (84 to 266 amino acid), and full length of *Sli*, in addition to the C-terminal region harboring the F-box protein interaction domain (114 to 393 amino acid) of the negative control gene (*RHC01H2G1617*) were constructed into the pGBKT7 vector [BD-*PP2*, BD-*Sli*, and BD-*StSLF(t)*]. Each type of AD-*S-RNase* was co-transformed with BD-*PP2*, BD-*Sli*, or BD-*StSLF(t)*, respectively, into the Y2H Gold Yeast Strain according to the manufacturer's instruction (Clontech). The yeast cell harboring pGADT7-T and pGBKT7–53 was used as a positive control, and the cell harboring pGADT7-T and pGBKT7-Lam was used as a negative control. The plate was then incubated for 4 days at 30 °C, and the yeast was grown on the synthetically defined medium (SD/-Trp/-Leu/-His/-Ade) as recorded. The experiments were repeated, independently, at least three times with similar results. The primers used in this analysis are listed in Supplementary Table 2.

**Luciferase complementation assay in *N. benthamiana* leaves**. The C-terminal region harboring the PP2 domain (85 to 266 amino acid) of *Sli* was cloned into pCAMBIA1301-cLUC (cLUC-*PP2*). The codon region of each *S-RNase* (S1-S14) except for its signal peptide was cloned into pCAMBIA1301-nLUC (nLUC-*S-RNase*). Each construct was next transformed into *Agrobacterium tumefaciens* strain GV3101. Each strain was grown at 28 °C in LB media with kanamycin (50 mg/L) and rifampicillin (50 mg/L) to an $OD_{600}$ of 0.8. Cells were then harvested by centrifugation at 3000 × g for 5 min and resuspended in 500 mM MES buffer containing 1 M $MgCl_2$ and 100 mM acetosyringone (Sigma), to a final $OD_{600}$ of 0.6. For co-infiltration, equal volumes of the strain harboring cLUC-*PP2* and the strain harboring nLUC-*S-RNase* were mixed and infiltrated into fully expanded leaves of 4-week-old tobacco using a needleless syringe. After the infiltration, plants were grown under dark for 8 h and then grown under 16 h light/8 h dark for 48 h. Then the substrate D-luciferin (Sigma) was sprayed onto the infiltrated tobacco leaf. After 5 min of incubation in dark, a chemiluminescence signal was observed and captured using NightSHADE LB 985 (Berthold, Germany). The experiments were repeated, independently, at least three times with similar results. The primers used in this analysis are listed in Supplementary Table 2.

**Reporting summary**. Further information on research design is available in the Nature Research Reporting Summary linked to this article.

## Data availability

The BSA-Seq data (SRR14637423 and SRR14637422) has been deposited in NCBI under project number PRJNA732509. All other data that support the findings of this study are available from the corresponding author upon reasonable request. Source data are provided with this paper.

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

## Acknowledgements

We thank Prof. K. Hosaka (Obihiro University of Agriculture and Veterinary Medicine), Prof. Jianjian Qi (Inner Mongolia University), Prof. Xueyong Yang (Institute of Vegetables and Flowers, Chinese Academy of Agricultural Sciences), and Prof. Guangtao Zhu (Yunnan Normal University) for experimental and bioinformatics support. This work was supported by the National Natural Science Foundation of China (32002032 and 31972433), Yunnan Science Fund (202005AE160015 and 2019FJ004 to Y.S.), Shenzhen Science and Technology Program (KQTD2016113010482651 to S.H.), and special funds for Science Technology Innovation and Industrial Development of Shenzhen Dapeng New District (RC201901-05 to S.H.). This work was also supported by Yunnan Provincial and Shenzhen Municipal Governments.

## Author contributions

Y.S., S.H., L.M., C.L., and C.Z. conceived and organized the research. Q.L., D.T., and Y.J. performed the bioinformatics analyses. B.Z., F.T., F.L., and H.Y. carried out the phenotyping assay. B.Z., F.T., F.L., M.G., and J.Z. performed the cloning and functional analysis of *Sli*. C.Z. and D.G. tested the efficiency of Sli in SI-breaking. C.Z., B.Z., F.T., X.L., and F.L. identified the alleles of S-RNases from diploid potatoes. Y.S., L.M., and S.H. wrote the manuscript. L.W. and H.G. revised the manuscript.

## Competing interests

The authors declare no competing interests.
