## [Peer Review File · Nature Communications]

REVIEWER COMMENTS

Reviewer #1 (Remarks to the Author):

The paper by Ma et al. reports isolation of the S-locus inhibitor (Sli) gene, which confers self-compatibility to otherwise self-incompatible diploid potato strains. They show that the gene underlying Sli, which they name 'NSF' (Non-S-locus F-box), encodes a pollen-expressed F-box protein that recognizes several different S-RNase protein variants, leading to their subsequent degradation via the ubiquitin/proteasome pathway. The research is a significant contribution as it elucidates the molecular mechanism of a well-known self-compatibility trait that has been promoted as a way to convert potato, a difficult to breed autotetraploid, into a diploid crop amenable to faster genetic improvement via conventional inbreds and F1 hybrid seed. The paper is also interesting as it describes a mutant F-box gene that apparently acquired pollen expression and recognizes a subset of stylar S-RNase proteins, unlike the S-locus F-box proteins that are normally expressed in pollen and specifically recognize only one or two S-RNases. A strong point of the paper is that it demonstrates that the NSF protein recognizes some of the many diverse S-RNase protein variants, but not others. While these interactions were assessed using in vitro methods (Y2H), which may not reflect what happens in vivo, they provide strong evidence that NSF/Sli recognizes S-RNase proteins. (They did not test for interactions with other proteins required for SI expression in the style).

Significant issues:

1. The research described in this paper is well thought out, thorough and of high quality. Unfortunately the same cannot be said of the written English in the manuscript text. While generally comprehensible, the text contains too many grammatical errors. The lack of line numbering makes it impractical to list all of these, which in any case should not be the responsibility of the reviewer. I strongly recommend having the manuscript thoroughly edited to improve grammar, word usage and sentence structure.
2. The authors fail to cite a recent similar work on Sli. On page 3, paragraph 2, the statement 'Unfortunately, the Sli gene has not been cloned yet.' is not entirely accurate. Eggers et al. (2020) reported cloning of Sli:
Eggers et al. (2020) The S-Locus Inhibitor gene encodes an F-box protein with a Lectin domain and crucially enables hybrid potato breeding. Solanaceae International Online Meeting. Granted, this was an oral presentation, but the abstract is available online (ftp://ftp.solgenomics.net/sgn_homepage/sol2020/InternationalSOL_Meeting2020_AbstractsBook20201108.pdf).
3. The introduction section leaves the impression that SI is the only factor preventing the breeding of potatoes at the diploid level. Another hurdle is inbreeding depression caused by genetic load, which is expected to be high in tetraploid potatoes, as well as SI diploids, due to their high level of heterozygosity. Converting the breeding system of cultivated potatoes from SI to SC requires only a mutation in one gene (Sli in this case, but S-RNase or SLF mutations would also suffice), whereas overcoming inbreeding depression and purging deleterious mutations could prove more challenging in the long run if they necessitate genome wide selection across multiple generations.
4. Page 4, paragraph 2. I do not understand why F1 PI 225689 x RH would produce a/a progeny (assuming A=Sli, and a=WT), unless the RH parent was used as female. If the PI accession was used as female (as implied by the way the cross is written), and if it is fully SI, then only Sli-bearing pollen should be compatible. This point needs to be clarified in the text. (Or I missed something.)
5. Only two transgenic diploid potato plants expressing the NSF (Sli) transgene were obtained, and the conclusions about the function of NSF are primarily based on the phenotypes of these two plants, both of which were SC. Identical NSF sequences were found in two other SC accessions (M6 and S. chacoense), supporting their conclusion that NSF is functionally the Sli gene. Nonetheless, the small number of primary transformants is a weakness of this paper. The authors state that it was difficult to obtain diploid primary transformants due to the high frequency of

tetraploid regenerates (presumably tetraploidy resulted from the transformation/regeneration conditions used, although this is not explained). Nonetheless, it would be very helpful if the authors could include data from additional transgenic plants, perhaps self-progeny of either of the two primary transgenic plants, if they have any such data. This would strengthen their functional evidence that NSF is Sli, and that it is both required and sufficient for SC in an otherwise SI genetic background.

6. The authors do not address the advantages and disadvantages of NSF/Sli based SC over other types of SC mutations. One potential drawback highlighted by this study is the fact that NSF does not interact with all S-RNases in vitro (the Y2H experiments). Thus the Sli gene may not convert SI to SC in all genetic backgrounds and S-haplotypes. This limitation contrasts with S-RNase loss of function mutations, which are known in other *Solanum* spp. (*S. pennellii*, *S. habrochaites* and *S. arcanum*) and have a stable SC phenotype because pollen retain the ability to recognize ALL non-self S-RNases. This results in an SC phenotype that is dominant over SI in heterozygotes with ANY S-haplotype (except the nonmutant S-haplotype in which an S-RNase LOF allele arose). Another limitation of NSF/Sli-based SC is that the SC mutation will segregate independently of the S-locus, which allows SC to revert to SI, for example in progeny of SI x Sli crosses. Thus from a breeding standpoint, NSF/Sli has potential limitations, based on the results of this study. The authors mention that S-RNase knockouts face the obstacle of lack of public acceptance of GMO crops, which is a valid point, however non-GMO LOF mutations have been found in natural populations of other SI species, and they may exist in diploid potatoes too.

7. Figure 1a. The pollen tubes are not easily visible, at least in the PDF version of the figure. It may be necessary to adjust the resolution or contrast of this figure. It would also help to add arrows or other symbols to indicate the position of pollen tube arrest within the styles.

8. Figure 4. The model shows NSF recognizing S2/S3 S-RNases in the pistil. I realize this is probably just for illustration purposes, but the Y2H and Luc assays show that S3 S-RNase is NOT recognized by NSF. If pollen bearing the NSF allele fail to recognize BOTH S-RNases in the pistil, the reaction will be incompatible. The model would be improved by clarifying what happens in both compatible and incompatible reactions involving NSF. Secondly, the model presents an alternative path to SC via gain-of-function mutation in an SLF that recognizes 'self' S-RNase. That's fine, but the model could be improved by incorporating another known route to SC based on loss-of-function mutations in the S-RNase (i.e. without a GOF mutation in either SLF or NSF). LOF mutations in the S-RNase are genetically simpler than GOF mutations and are the predominant route to SC, at least in the wild tomatoes, which are closely related to potato. I appreciate that Sli is more analogous to an SLF GOF mutation, since both occur in pollen, but the model would be more complete if it also included S-RNase mutations.

9. Figure 2. The numbers on the pie chart (Fig 2b) are said to represent the numbers of 'species that transfer from SI to SC after crossing with RH...'. However, each slice of the pie represents a particular species or subspecies, so I'm guessing the numbers actually represent accessions within species/subspecies. In any case the numbers don't mean much without also knowing the total number of lines that were tested within each group. In the text (page 5), '110 of the 125 species showed SC', however it is not clear to which species the 15 accessions (?) that did not convert to SC belonged. Were they all in one species, or a few in each group? Also, on Fig 2c, the pollen and style samples presumably come from mature flowers, but this is not explicitly stated in the legend.

10. The author's use of the name 'NSF' (non S-locus F-box protein) instead of, or in addition to 'Sli' creates confusion by unnecessarily adding alternative terminology to the literature. The gene symbol Sli is well established, has precedence, and reflects the SC phenotype. Furthermore, there are MANY F-box encoding genes in the genome, most of which are NOT at the S-locus. Thus the name 'Non S-locus F-box' could conceivably be understood to refer to many other F-box genes, which might create some confusion. Why not stick with Sli?

Minor issues:

-page 2, paragraph 1: replace 'exacerbating' with 'complicating', 'impeding' or a similar term
-p2, par2: use 'true seed' or 'botanical seed' to distinguish from 'seed potatoes' (i.e. tubers).

- p2, par3: define 'SLF' at first use (i.e. S-locus F-box)
- p2, par3 (and elsewhere): italicize genes or mRNAs, not gene products or proteins.
- p3, par1: loss of S-RNase function has been documented not just through knock-out mutations, but also from amino acid substitutions that eliminate RNase activity.
- p3, par1: replace 'degradation of this gene' by 'degradation of this protein'
- p3, last paragraph: the Sli gene acts gametophytically, thus it is not entirely accurate to refer to it as 'dominant'. Suggest inserting 'or gametophytic factor' after 'dominant gene'. Also, the authors should clarify the direction of the initial cross (I assume PI female x RH male).
- p4, par2: the chi-square statistic should be based on an expected 1:2:1 segregation, although it doesn't hurt to also compute the goodness-of-fit to the 1:1, as shown.
- p4: replace 'narrowe' with 'narrow'
- supplementary figure 2b, d, f and g: explain what M1 and M2 represent. Also the figure would be improved by having nontransformed diploid and tetraploid samples as references. The empty vector control has presumably gone through transformation and regeneration also, so could easily have undergone a change in ploidy.
- supplementary figure 3. The figure shows that the sequence of NSF from RH is 100% identical to that found in *S. chacoense* or M6. Therefore the figure could be simplified by presenting just the sequence of RH, and in the legend stating that the other two sequences were identical.

Reviewer #2 (Remarks to the Author):

I am glad to have a chance to review the manuscript, entitled "A non S-locus F-box gene breaks self-incompatibility in diploid potatoes. A diploid hybrid breeding has now become a hot topic for potato breeders. The well-known self-compatibility-inducing gene Sli has opened the way to develop diploid inbred lines, but this gene has not been isolated and its molecular function has remained unknown. This article discovered that a responsible gene is PGSC0003DMG400016861, a F-box protein gene with the F-box domain and PP2 domain and that it works like a general S-RNase inhibitor. This is certainly worth to be published.

There are two major concerns on this article.

The authors identified a NSF gene that imparted SC to RH. The genomic region containing the NSF has been supposed to be a candidate region for the Sli gene (Clot et al. 2020). The sequence similarity of this region is very high not only between RH and M6 but also between these and several famous cultivars (Clot et al. 2020). This similarity has been demonstrated in Supplementary Fig. 3 where 100% similarity among SC *S. chacoense*, M6 and RH is shown in the sequence of the NSF gene (although I can't understand very well about this sequence because start codon and stop codon are not indicated). These indicate that the allelic variation in the NFS gene sequence is not so important because similar sequence can be found in many genotypes irrespective of being SC or SI. More important could be the presence or absence of a 536 bp insertion in the promoter region. SC *S. chacoense*, M6 and RH have the same size insertion present at the same position? The comparison of the promoter region is highly requested.

The second major concern is on the genetic action of the NSF/Sli gene. The authors described, "Thus, only the pollen harboring the SC gene can penetrate the self style and fulfil fertilization to produce progeny, and all the F2 progeny would carry the SC gene and exhibit self-compatible phenotype". So, they suggested that the NSF/Sli is gametophytically expressed. In contrast, a sporophytic action for the Sli gene has been proposed by Hosaka and Hanneman (1998). Based on sporophytic action hypothesis, pollen grains produced from a Sli-carrying pollen parent are all compatible in self-crossing. Thus, from a cross between SC (Sli/-) and SI (-/-), the progeny plants are segregated to a ratio of 1 SC (Sli/-) : 1 SI (-/-). Self-crossing of the SC (Sli/-) results in a ratio of 1 Sli/Sli : 2 Sli/- : 1 -/-; however, Sli/Sli was presumed to be absent due to a tight linkage with lethality (Hosaka and Hanneman 1998). Based on the sporophytic action hypothesis, the F2 population in Fig. 1d would be segregated to a ratio of 2 SC : 1 SI. If the gametophytic action is employed, only pollen grains carrying NSF/Sli are fertilized and generated only SC F2 plants. In this context, expression of NSF/Sli in pollen grains would be segregated to a ratio of 1:1; a half pollen grains would produce mRNA in in situ mRNA hybridization of NSF in Fig. 2d. in situ mRNA hybridization of NSF is suggested to conduct not with maturing pollen grains in the anther but with mature, fertile pollen grains released from the anther. Please verify which is a likely action of

NSF/Sli.

Minor comments

Under the heading "SC gene colocalizes with a segregation distortion region in RH" in Results, a segregation ratio of A/A:A/a=3030:2954≈1:1 was observed for the F2 progeny from crossing PI 225689 and RH. A/A and A/a genotypes were determined by marker analysis? Why did the authors speculate that A/a is a heterozygote SC and a/a is a homozygote SI plant? This is a question relating to the major concern described above. In the following paragraph, the authors described, "In total, 6624 F2 individuals produced from the 131 selfed F1 plants were used to narrow down the SC gene to a 64.61 kb interval (Fig. 1c)". From this sentence I thought that SC and SI were segregated among F2 individuals, so that the candidate region could be narrowed down. So, not all F2 individuals were SC. If my understanding is correct, A/a (SC) and a/a (SI) were identified in the F2 population and a sporophytic action should be proposed.

Under the heading "NSF is efficient in rendering SC to SI conversion in diploid potatoes" in Results, it is described that 125 SI diploid species were pollinated. apparently, 125 is not the number of species but probably the number of accessions or clones, plants, etc. Please be clarified. In addition, *S. tuberosum* ssp. *andigena* and ssp. *tuberosum* are tetraploid and self-fertile. Did the authors use dihaploid genotypes of these species? In the same paragraph, the sentence "The unchanged SI phenotype of 15 diploid lines after the crossing may be caused by unsuccessful pollination as the flowering time of these lines were different from that of RH or E172." is not clear. I assume F1 hybrids were investigated for self-compatibility. Does this sentence mean that the 15 diploid lines were not true hybrids, so that they were not SC?

I think Fig. 4 is not necessary.

In Methods, the affiliation of Prof. K. Hosaka is incorrect (not "Obihiro University" but "Obihiro University of Agriculture and Veterinary Medicine"). In the same sentence, what kind of genotypes for *Solanum chacoense* (clonal or accession ID) was provided from Prof. Hosaka?

In Methods, the heading "Polygenetic analysis of NSF/Sli" should be changed to "Phylogenetic analysis of NSF/Sli".

Reference 20. "Hosaka, K. & E. Hanneman, Jr., R." should be changed to "Hosaka, K. & Hanneman, Jr., R. E.".

Reviewer #3 (Remarks to the Author):

This manuscript is presented to demonstrate that a cloned gene (NSF) is the contributing factor to over-coming self-incompatibility (SI) in diploid potato. They provide fine-mapping, pollen tube growth, quantitative RT-PCR, in situ hybridization, plant transformation and yeast 2-hybrid assay to support their work. I think the authors may have cloned a gene that contributes to overcoming self-compatibility (SC) but they are not providing seed set in the transgenic plants to support their results. Our work in SI/SC tells us that fruit set does not always correlate with seed set. We also use stilar squashes to look at pollen tube growth. In a number of instances, we see pollen tube growth to the ovules but no seed set.

Another point to address is the phenotyping of plant for SI/SC. There is a great environmental influence on this phenotype. The authors' phenotyping methods lack detail and create doubt in the results.

I question why the authors had to generated over 3,000 transgenic events. Explanation would be valuable. I am also concerned why so many tetraploid events are recovered in the 26 events studied.

The plant material used was poorly detailed. Furthermore, Clot et al. (2020) provided a means to

genotype the Sli region in potato lines. I feel knowing that genotyping would add confidence in the correct interpretation to the results observed. I also feel that the authors should more clearly address their work in the introduction. In the results the authors refer to markers M-1 and M-2. These markers are not referenced. Are they based upon Clot et al.?

The authors present results of using RH to overcome SI in a diverse set of germplasm (Fig 2b and Suppl Table 1). I think this work is not critical to the paper. Furthermore, our experience in using RH to transmit SC is mixed. We rarely if any see a 1:1 segregation for SC/SI. We would not classify RH as a primary clone to overcome SC (SI-breaker).

According to Kubo et al (2015. PMID: 27246052) there are 14 pollen-specific SLFs in potato. Why this number disagrees with the authors' results and discussion?

The authors provide some evolutionary conjecture in the discussion. This model is weak and does not need to be in the manuscript.

BSA-Seq analysis was based on 80 progeny. The number of progeny used is much larger than other previously published studies. Why so many progeny used?

My suggestion is that the authors resubmit when they can provide seed set data to support the transgenic results and address the issues presented above.

Reviewer #1 (Remarks to the Author):

The paper by Ma et al. reports isolation of the *S-locus* inhibitor (*Sli*) gene, which confers self-compatibility to otherwise self-incompatible diploid potato strains. They show that the gene underlying *Sli*, which they name 'NSF' (Non-S-locus F-box), encodes a pollen-expressed F-box protein that recognizes several different S-RNase protein variants, leading to their subsequent degradation via the ubiquitin/proteasome pathway. The research is a significant contribution as it elucidates the molecular mechanism of a well-known self-compatibility trait that has been promoted as a way to convert potato, a difficult to breed autotetraploid, into a diploid crop amenable to faster genetic improvement via conventional inbreds and F₁ hybrid seed. The paper is also interesting as it describes a mutant F-box gene that apparently acquired pollen expression and recognizes a subset of stylar S-RNase proteins, unlike the S-locus F-box proteins that are normally expressed in pollen and specifically recognize only one or two S-RNases. A strong point of the paper is that it demonstrates that the NSF protein recognizes some of the many diverse S-RNase protein variants, but not others. While these interactions were assessed using *in vitro* methods (Y2H), which may not reflect what happens *in vivo*, they provide strong evidence that NSF/*Sli* recognizes S-RNase proteins. (They did not test for interactions with other proteins required for SI expression in the style).

Response:

We thank Reviewer for this positive comment.

Significant issues:

1. The research described in this paper is well thought out, thorough and of high quality. Unfortunately the same cannot be said of the written English in the manuscript text. While generally comprehensible, the text contains too many grammatical errors. The lack of line numbering makes it impractical to list all of these, which in any case should not be the responsibility of the reviewer. I strongly recommend having the manuscript thoroughly edited to improve grammar, word usage and sentence structure.

Response:

We feel very sorry to make many mistakes in preparing the manuscript. In the revised version, the maintext has been carefully edited by consultants from EditSprings (www.editsprings.com/en/).

2. The authors fail to cite a recent similar work on *Sli*. On page 3, paragraph 2, the statement 'Unfortunately, the *Sli* gene has not been cloned yet.' is not entirely accurate. Eggers et al. (2020) reported cloning of *Sli*: Eggers et al. (2020) The S-Locus Inhibitor gene encodes an F-box protein with a Lectin domain and crucially enables hybrid potato breeding. Solanaceae International Online Meeting. Granted, this was an oral presentation, but the abstract is available

online(ftp://ftp.solgenomics.net/sgn_homepage/sol2020/InternationalSOL_Meeting2020_AbstactsBook20201108.pdf).

Response:

We thank Reviewer for this suggestion. The statement was modified and the recent work was cited in the revised manuscript (Page 3, line 13-15). We submitted this manuscript together with this group, and planned to make a back-to-back publication.

3. The introduction section leaves the impression that SI is the only factor preventing the breeding of potatoes at the diploid level. Another hurdle is inbreeding depression caused by genetic load, which is expected to be high in tetraploid potatoes, as well as SI diploids, due to their high level of heterozygosity. Converting the breeding system of cultivated potatoes from SI to SC requires only a mutation in one gene (Sli in this case, but S-RNase or SLF mutations would also suffice), whereas overcoming inbreeding depression and purging deleterious mutations could prove more challenging in the long run if they necessitate genome wide selection across multiple generations.

Response:

We thank Reviewer for this suggestion and have modified the introduction section by adding another big hurdle (inbreeding depression) in diploid hybrid potato breeding. (Page 2, line 29).

4. Page 4, paragraph 2. I do not understand why F₁ PI 225689 x RH would produce a/a progeny (assuming A=Sli, and a=WT), unless the RH parent was used as female. If the PI accession was used as female (as implied by the way the cross is written), and if it is fully SI, then only Sli-bearing pollen should be compatible. This point needs to be clarified in the text. (Or I missed something.)

Response:

Although PI 225689 is a fully SI line (a/a type), it is possible that the *S-locus* F-box proteins harbored by RH pollen, not the *Sli* allele (A), can recognize and detoxify the S-RNase expressed in PI 225689. Thus, the a/a progeny was produced from RH crossed with PI 225689.

5. Only two transgenic diploid potato plants expressing the NSF (Sli) transgene were obtained, and the conclusions about the function of NSF are primarily based on the phenotypes of these two plants, both of which were SC. Identical NSF sequences were found in two other SC accessions (M6 and S. chacoense), supporting their conclusion that NSF is functionally the Sli gene. Nonetheless, the small number of primary transformants is a weakness of this paper. The authors state that it was difficult to obtain diploid primary transformants due to the high frequency of tetraploid regenerates (presumably tetraploidy resulted from the transformation/regeneration conditions used, although this is not explained). Nonetheless, it would be very helpful if the authors

could include data from additional transgenic plants, perhaps self-progeny of either of the two primary transgenic plants, if they have any such data. This would strengthen their functional evidence that NSF is Sli, and that it is both required and sufficient for SC in an otherwise SI genetic background.

Response:

Currently, gene transformation rate to produce positive diploid lines is low, as 24 positive transgenic lines were changed to tetraploids. This unexpected alteration may occur during regeneration process, since this process tends to induce genome instability¹. We did the self-progeny of these two transgenic lines. Each line produced a lot of seeds after selfing. This information has been added to Fig. 1g to strengthen the functional evidence of *NSF/Sli* as suggested. We thank Reviewer this suggestion.

Fig. 1g was modified to show status of self-progeny of the two *NSF/Sli* transgenic diploid lines.

6. The authors do not address the advantages and disadvantages of NSF/Sli based SC over other types of SC mutations. One potential drawback highlighted by this study is the fact that NSF does not interact with all S-RNases in vitro (the Y2H experiments). Thus the Sli gene may not convert SI to SC in all genetic backgrounds and S-haplotypes. This limitation contrasts with S-RNase loss of function mutations, which are known in other *Solanum* spp. (*S. pennellii*, *S. habrochaites* and *S. arcanum*) and have a stable SC phenotype because pollen retain the ability to recognize ALL non-self S-RNases. This results in an SC phenotype that is dominant over SI in heterozygotes with ANY S-haplotype (except the nonmutant S-haplotype in which an S-RNase LOF allele arose). Another limitation of NSF/Sli-based SC is that the SC mutation will segregate independently of the S-locus, which allows SC to revert to SI, for example in progeny of SI x Sli crosses. Thus from

a breeding standpoint, NSF/Sli has potential limitations, based on the results of this study. The authors mention that S-RNase knockouts face the obstacle of lack of public acceptance of GMO crops, which is a valid point, however non-GMO LOF mutations have been found in natural populations of other SI species, and they may exist in diploid potatoes too.

Response:

We thank Reviewer for this suggestion. The limitation of using NSF/Sli to break SI and the potential application of the non-GMO LOF S-RNase in diploid potatoes have been added to the maintext (Page 7 line 4-14). As Reviewer mentioned, we found the non-GMO S-RNase LOF mutations do exist in natural potato populations, such as PG6359². Since *Sli* is located on Ch12 and *S-RNase* is on Ch01, usage of SC genes from different sources in breeding can effectively avoid future genetic bottlenecks at these loci, considering that *S-RNase* is near centromere while *Sli* is near telomere.

7. Figure 1a. The pollen tubes are not easily visible, at least in the PDF version of the figure. It may be necessary to adjust the resolution or contrast of this figure. It would also help to add arrows or other symbols to indicate the position of pollen tube arrest within the styles.

Response:

The figure has been replaced by a high resolution one and arrow is used to indicate the arrest of pollen tube as suggested. We thank Reviewer for this suggestion.

8. Figure 4. The model shows NSF recognizing S2/S3 S-RNases in the pistil. I realize this is probably just for illustration purposes, but the Y2H and Luc assays show that S3 S-RNase is NOT recognized by NSF. If pollen bearing the NSF allele fail to recognize BOTH S-RNases in the pistil, the reaction will be incompatible. The model would be improved by clarifying what happens in both compatible and incompatible reactions involving NSF. Secondly, the model presents an alternative path to SC via gain-of-function mutation in an SLF that recognizes 'self' S-RNase. That's fine, but the model could be improved by incorporating another known route to SC based on loss-of-function mutations in the S-RNase (i.e. without a GOF mutation in either SLF or NSF). LOF mutations in the S-RNase are genetically simpler than GOF mutations and are the predominant route to SC, at least in the wild tomatoes, which are closely related to potato. I appreciate that Sli is more analogous to an SLF GOF mutation, since both occur in pollen, but the model would be more complete if it also included S-RNase mutations.

Response:

We thank Reviewer for this suggestion. The model was modified to show both compatible and incompatible reactions involving NSF/Sli. The SI to SC transition caused by LOF mutations of the S-RNase was also presented as suggested.

The model is modified to show loss of function mutations of *S-RNases* to confer SC as suggested.

9. Figure 2. The numbers on the pie chart (Fig 2b) are said to represent the numbers of 'species that transfer from SI to SC after crossing with RH...'. However, each slice of the pie represents a particular species or subspecies, so I'm guessing the numbers actually represent accessions within species/subspecies. In any case the numbers don't mean much without also knowing the total number of lines that were tested within each group. In the text (page 5), '110 of the 125 species showed SC', however it is not clear to which species the 15 accessions (?) that did not convert to SC belonged. Were they all in one species, or a few in each group? Also, on Fig 2c, the pollen and style samples presumably come from mature flowers, but this is not explicitly stated in the legend.

Response:

We thank Reviewer for this correction and feel very sorry for this mistake. It is 'accession', not 'species'. We have modified the wording in the revised manuscript. The total number we used in this experiment is 125 lines, and 15 lines failed to cross with RH or E172. The species information

of the 15 exceptions was provided in Supplementary Table 1. The total number of accessions within each species was also added to the maintext (Page 5, line 27-32). We modified the legend of Fig. 2c to describe the different stages of pollen and tissue used in the qPCR analysis (Page 17, line 11-13).

10. The author's use of the name 'NSF' (non S-locus F-box protein) instead of, or in addition to 'Sli' creates confusion by unnecessarily adding alternative terminology to the literature. The gene symbol Sli is well established, has precedence, and reflects the SC phenotype. Furthermore, there are MANY F-box encoding genes in the genome, most of which are NOT at the S-locus. Thus the name 'Non S-locus F-box' could conceivably be understood to refer to many other F-box genes, which might create some confusion. Why not stick with Sli?

Response:

We do agree that 'NSF' creates confusion. *NSF* was used instead of using *Sli* mainly based on two reasons. First, a group from Wageningen was using the wild diploid potato species, *Solanum chacoense* Bitt., to clone *Sli* gene, we were using RH as a starting material. Thus, NSF was used to differentiate two studies, indicating two group applied different strategies or routes to clone *Sli*. Another reason is we didn't know whether the SC gene (*NSF*) from RH is the same to *Sli* from *Solanum chacoense* Bitt., only after we confirmed that the coding sequences of *NSF* and *Sli* are exactly the same. We changed '*NSF/Sli*' to '*Sli*' to avoid this confusion after two genes were confirmed to be the same gene.

Minor issues:

-page 2, paragraph 1: replace 'exacerbating' with 'complicating','impeding' or a similar term

Response:

OK (Page 2, line 18).

-p2, par2: use 'true seed' or 'botanical seed' to distinguish from 'seed potatoes' (i.e. tubers).

Response:

OK (Page 2, line 26).

-p2, par3: define 'SLF' at first use (i.e. S-locus F-box)

Response:

OK (Page 2, line 34).

-p2, par3 (and elsewhere): italicize genes or mRNAs, not gene products or proteins.

Response:

OK.

-p3, par1: loss of S-RNase function has been documented not just through knock-out mutations, but also from amino acid substitutions that eliminate RNase activity.

Response:

OK. We used 'loss of function mutation of *S-RNase*' to replace 'knock out of *S-RNase*' (Page 3, line 5).

-p3, par1: replace 'degradation of this gene' by 'degradation of this protein'

Response:

OK (Page 3, line 6).

-p3, last paragraph: the *Sli* gene acts gametophytically, thus it is not entirely accurate to refer to it as 'dominant'. Suggest inserting 'or gametophytic factor' after 'dominant gene'. Also, the authors should clarify the direction of the initial cross (I assume PI female x RH male).

Response:

OK (Page 3, line 32; Page 4, line 2).

-p4, par2: the chi-square statistic should be based on an expected 1:2:1 segregation, although it doesn't hurt to also compute the goodness-of-fit to the 1:1, as shown.

Response:

As mentioned below, our genetic results have proved the gametophytic action of *Sli*. Thus, we calculated chi-square statistic based on the 1:1 segregation, not 1:2:1 segregation.

-p4: replace 'narrowe' with 'narrow'

Response:

OK (Page 4, line 28).

-supplementary figure 2b, d, f and g: explain what M1 and M2 represent. Also the figure would be improved by having nontransformed diploid and tetraploid samples as references. The empty vector control has presumably gone through transformation and regeneration also, so could easily have undergone a change in ploidy.

Response:

OK.

-supplementary figure 3. The figure shows that the sequence of NSF from RH is 100% identical to that found in *S. chacoense* or M6. Therefore the figure could be simplified by presenting just the sequence of RH, and in the legend stating that the other two sequences were identical.

Response:

OK. As suggested by other reviewer, we provide a sequencing alignment not only the coding region, but also the promoter region among RH, M6, and E172. This figure is updated in the revised manuscript.

Reference:

1. Fossi, M. *et al.* Regeneration of *Solanum Tuberosum* Plants from Protoplasts Induces Widespread Genome Instability. *Plant Physiol.* 2019, 180 (1), 78–86.
2. Zhang, C. *et al.* The genetic basis of inbreeding depression in potato. *Nat. Genet.* 51, 374–378 (2019).

Reviewer #2 (Remarks to the Author):

I am glad to have a chance to review the manuscript, entitled “A non S-locus F-box gene breaks self-incompatibility in diploid potatoes. A diploid hybrid breeding has now become a hot topic for potato breeders. The well-known self-compatibility-inducing gene *Sli* has opened the way to develop diploid inbred lines, but this gene has not been isolated and its molecular function has remained unknown. This article discovered that a responsible gene is PGSC0003DMG400016861, a F-box protein gene with the F-box domain and PP2 domain and that it works like a general S-RNase inhibitor. This is certainly worth to be published.

Response:

We thank Reviewer for this positive comment.

There are two major concerns on this article.

The authors identified a NSF gene that imparted SC to RH. The genomic region containing the NSF has been supposed to be a candidate region for the Sli gene (Clot et al. 2020). The sequence similarity of this region is very high not only between RH and M6 but also between these and several famous cultivars (Clot et al. 2020). This similarity has been demonstrated in Supplementary Fig. 3 where 100% similarity among SC *S. chacoense*, M6 and RH is shown in the sequence of the NSF gene (although I can't understand very well about this sequence because start codon and stop codon are not indicated). These indicate that the allelic variation in the NFS gene sequence is not so important because similar sequence can be found in many genotypes irrespective of being SC or SI. More important could be the presence or absence of a 536 bp insertion in the promoter region. SC *S. chacoense*, M6 and RH have the same size insertion present at the same position? The comparison of the promoter region is highly requested.

Response:

We thank Reviewer for this suggestion. The comparison of the promoter region of NSF/Sli among *S. chacoense*, M6 and RH was provided. All the SC lines have an insertion at the promoter region of *Sli* (Page 5, line 19-25). The start and stop codon are also indicated in Supplementary Fig. 3.

The second major concern is on the genetic action of the NSF/Sli gene. The authors described, “Thus, only the pollen harboring the SC gene can penetrate the self style and fulfil fertilization to produce progeny, and all the F₂ progeny would carry the SC gene and exhibit self-compatible phenotype”. So, they suggested that the NSF/Sli is gametophytically expressed. In contrast, a sporophytic action for the Sli gene has been proposed by Hosaka and Hanneman (1998). Based on sporophytic action hypothesis, pollen grains produced from a Sli-carrying pollen parent are all compatible in self-crossing. Thus, from a cross between SC (Sli/-) and SI (-/-), the progeny plants are segregated to a ratio of 1 SC (Sli/-) : 1 SI (-/-). Self-crossing of the SC (Sli/-) results in a ratio of 1 Sli/Sli : 2 Sli/- : 1 -/-; however, Sli/Sli was presumed to be absent due to a tight linkage with lethality (Hosaka and Hanneman 1998). Based on the sporophytic action hypothesis, the F₂ population in Fig. 1d would be segregated to a ratio of 2 SC : 1 SI. If the gametophytic action is employed, only pollen grains carrying NSF/Sli are fertilized and generated only SC F₂ plants. In this context, expression of NSF/Sli in pollen grains would be segregated to a ratio of 1:1; a half pollen grains would produce mRNA in in situ mRNA hybridization of NSF in Fig. 2d. in situ mRNA hybridization of NSF is suggested to conduct not with maturing pollen grains in the anther but with mature, fertile pollen grains released from the anther. Please verify which is a likely action of NSF/Sli.

Response:

We thank Reviewer for this suggestion. According to the genetic analysis, we believe the gametophytic action should be correct for *Sli*. First, the genotype (*Sli/Sli*) was detected in the F₂ progeny or the S₁ population of RH. Thus, at least in our materials, *Sli* is not tightly linked with a

lethal gene. Second, an extreme segregation distortion phenomenon (1 *Sli/Sli*: 1 *Sli*/- : 0 -/-) was observed in F₂ progeny from two F₁ populations (PI225689×RH and RH×Y16-90) and one S₁ population of RH. In either the F₂ or the S₁ populations, no SI genotype (-/-) was identified. Third, pollen of RH was used as *Sli* donor by Dr. Chunzhi Zhang to pollinate 28 SI lines to create SC F₁ individuals (Supplementary Table 1). After generations of selfing of these SC plants in creation of diploid inbred lines, she didn't observe SC progeny return back to SI, indicating the genotype (-/-) should no longer exist in these populations. In addition, the group from Wageningen university independently discovered this segregation distortion phenomenon in their populations too. Thus, we concluded that the SD in the F₂ population and the S₁ population of RH are caused by the gametophytic SI. On the other hand, the sporophytic action hypothesis reported in the Hosaka and Hanneman paper was based on segregation of the SC/SI phenotype in F₁ and F₂ populations, not based on segregation of the genotype. As SC/SI phenotype would be affected by uncontrollable environmental factors and male sterility, it is possible that segregation rate of the SC/SI phenotype is different from that of the genotype.

For Fig. 2d, our main purpose is to show *Sli* is highly expressed in the mature pollen tissue. Consistent to our expectations, mature pollen has the highest express of *Sli*, and thus was used in *in situ* mRNA hybridization (d4). We agree with Reviewer that a half of pollen grains would produce *Sli* mRNA in *in situ* mRNA hybridization of *Sli*. However, the *in situ* mRNA hybridization may not be a good system to reflect this fact, as a slice of sample revealed in Fig. 2d only represents a very small fragment of the mature pollen tissue. Actually, we did observe the mature pollens that didn't express *Sli* in the *in situ* experiment. Thus, we think genetic analysis is more reliable than the *in situ* mRNA hybridization experiment to show the gametophytic action of *Sli*.

In situ mRNA hybridization of *NSF/Sli* in the mature stamen. Pollens that did not express *Sli* is indicated by the arrows.

Minor comments

Under the heading “SC gene colocalizes with a segregation distortion region in RH” in Results, a segregation ratio of $A/A:A/a=3030:2954\approx 1:1$ was observed for the F₂ progeny from crossing PI 225689 and RH. *A/A* and *A/a* genotypes were determined by marker analysis? Why did the authors speculate that *A/a* is a heterozygote SC and *a/a* is a homozygote SI plant? This is a question relating to the major concern described above. In the following paragraph, the authors described, “In total, 6624 F₂ individuals produced from the 131 selfed F₁ plants were used to narrow down the SC gene to a 64.61 kb interval (Fig. 1c)”. From this sentence I thought that SC and SI were segregated among F₂ individuals, so that the candidate region could be narrowed down. So, not all F₂ individuals were SC. If my understanding is correct, *A/a* (SC) and *a/a* (SI) were identified in the F₂ population and a sporophytic action should be proposed.

Response:

Firstly, the genotype segregation ratio of $A/A:A/a=3030:2954\approx 1:1$ was determined by the marker analysis. The state that ‘*A/a* is a heterozygote SC and *a/a* is a homozygote SI plant’ is mostly based on BSA-Sequence and genetic analysis of F₁ population. BSA-Sequence indicated that the SC phenotype is controlled by only one locus at the end of Ch12 (Fig. 1b), which can be represented by letter ‘A’ (dominance) or ‘a’ (recessiveness). $SC:SI=1:1$ in F₁ progeny indicates that one parent is heterozygote while the other is homozygote in this SC controlling locus. Resequencing data of the two parents, PI225689 and RH, indicates that the SC controlling locus is heterozygote in RH and homozygote in PI225689. Thus, RH is SC, and SC is dominant (A) and SI is recessive (a). The state that ‘all the F₂ progeny would carry the SC gene and exhibit self-compatible phenotype’ is based on genotype analysis of F₂ progeny that show an extreme segregation distortion. We proposed a gametophytic action of *Sli* for this extreme segregation distortion by stating that ‘only the pollen harboring the SC gene can penetrate the self-style and fulfil fertilization to produce progeny’. As all F₂ plants contain SC gene, there is no need to phenotype these F₂ individuals and we just did genotype of these plants to further narrow down the target gene.

Under the heading “NSF is efficient in rendering SC to SI conversion in diploid potatoes” in Results, it is described that 125 SI diploid species were pollinated. apparently, 125 is not the number of species but probably the number of accessions or clones, plants, etc. Please be clarified. In addition, *S. tuberosum* ssp. *andigena* and ssp. *tuberosum* are tetraploid and self-fertile. Did the authors use dihaploid genotypes of these species? In the same paragraph, the sentence “The unchanged SI phenotype of 15 diploid lines after the crossing may be caused by unsuccessful pollination as the flowering time of these lines were different from that of RH or E172.” is not

clear. I assume F1 hybrids were investigated for self-compatibility. Does this sentence mean that the 15 diploid lines were not true hybrids, so that they were not SC?

Response:

We thank Reviewer for this suggestion. It is ‘accession’, not ‘species’. We have modified the wording in the revised manuscript. 15 lines failed to cross with RH or E172. As all the lines including RH were planted at the same time, but their flowering time are different. Thus, the failure to set fruit of 15 lines mostly be caused by unsuccessful pollination. We rewrite this sentence to avoid confusion (Page 5, line 27-32). As Reviewer suggested, *S. tuberosum* ssp. andigena and ssp. tuberosum are tetraploid and self-fertile. When these accessions were introduced from CIP, the remark information of these materials indicate that they were classified as *S. tuberosum* ssp. andigena or ssp. Tuberosum. However, after resequencing of these materials (Qi et al, unpublished data) showed that these materials are actually diploids.

I think Fig. 4 is not necessary.

Response:

As appearance of *Sli* represents a new way to confer SC to SI plants, which is different from the gain of function mutations of SLFs or loss of function mutations of S-RNase, we think Fig. 4 is a new complement to the two-step model. We add more information to this Figure as suggested by Reviewer 1. If Reviewer insists to remove this figure, we could move it to Supplementary Figures.

The model is modified to show loss of function mutations of *S-RNases* to confer SC as suggested.

In Methods, the affiliation of Prof. K. Hosaka is incorrect (not “Obihiro University” but “Obihiro University of Agriculture and Veterinary Medicine”). In the same sentence, what kind of genotypes for *Solanum chacoense* (clonal or accession ID) was provided from Prof. Hosaka?

Response:

We thank Reviewer for this correction. The affiliation of Prof. K. Hosaka is corrected and the SC material chc525-3 was kindly provided from Prof. Hosaka. We selected E172 from a vigorous and SC clone in the selfed progeny of chc525-3 to cross with E (Page 8, line 2-4). We really appreciate for the great contributions of Prof. K. Hosaka to this project (Page 14, line 1).

Cover of the letter kindly sent from Prof. K. Hosaka

In Methods, the heading “Polygenetic analysis of NSF/Slu” should be changed to “Phylogenetic analysis of NSF/Slu”.

Response:

OK.

Reference 20. “Hosaka, K. & E. Hanneman, Jr., R.” should be changed to “Hosaka, K. & Hanneman, Jr., R. E.”.

Response:

OK.

Reviewer #3 (Remarks to the Author):

This manuscript is presented to demonstrate that a cloned gene (NSF) is the contributing factor to over-coming self-incompatibility (SI) in diploid potato. They provide fine-mapping, pollen tube

growth, quantitative RT-PCR, in situ hybridization, plant transformation and yeast 2-hybrid assay to support their work.

Response:

We thank Reviewer for this positive comment.

I think the authors may have cloned a gene that contributes to overcoming self-compatibility (SC) but they are not providing seed set in the transgenic plants to support their results. Our work in SI/SC tells us that fruit set does not always correlate with seed set. We also use stylar squashes to look at pollen tube growth. In a number of instances, we see pollen tube growth to the ovules but no seed set.

Response:

We thank Reviewer for this suggestion. We did the self-progeny of these two *Sli* transgenic lines. Each line produced a lot of seeds after selfing. This information has been added to Fig. 1g to strengthen the functional evidence of *NSF/Sli* as suggested.

Fig. 1g was modified to show the status of self-progeny of the two *NSF/Sli* transgenic diploid lines.

Another point to address is the phenotyping of plant for SI/SC. There is a great environmental influence on this phenotype. The authors' phenotyping methods lack detail and create doubt in the results.

Response:

We thank Reviewer for this suggestion. Three clones for each genotype were planted, and >30 self-pollinations were carried out for each clone. The genotype was considered as SC plant when ≥ 2 fruits were set on all the three clones; considered as SI plant when no fruit was set on all the three clones; considered as UI (Uncertain) plant when ≤ 1 fruit was set on each clone. This detail was added in the revised maintext (Page 8, line 15 to 18).

I question why the authors had to generated over 3,000 transgenic events. Explanation would be valuable. I am also concerned why so many tetraploid events are recovered in the 26 events studied.

Response:

Currently, gene transformation rate to produce positive diploid lines is low, as 24 positive transgenic lines were changed to tetraploids. This unexpected alteration may occur during regeneration process, since this process tends to induce genome instability¹. Thus, we have to enlarge the number of explants used in the *Sli* transformation to make sure we could get positive diploid lines.

The plant material used was poorly detailed. Furthermore, Clot et al. (2020) provided a means to genotype the *Sli* region in potato lines. I feel knowing that genotyping would add confidence in the correct interpretation to the results observed. I also feel that the authors should more clearly address their work in the introduction. In the results the authors refer to markers M-1 and M-2. These markers are not referenced. Are they based upon Clot et al.?

Response:

We thank Reviewer for this suggestion. We provide more details about the plant materials kindly shared by Prof. K. Hosaka (Page 8, line 2-4). The paper (Clot et al, 2020) have been introduced and cited in introduction (Page 3, line 19-20). The two markers, M-1 and M-2, were designed based on the BSA-sequencing and resequencing data in 2017, not based upon the results of Clot *et al.* The peak of SNP index (Δ) of SC and SI pools in BSA-sequencing data showed the physical position of SC locus in RH. Two InDel markers (named as M-1 and M-2) that were heterozygous in RH and homozygous in PI 225689 were designed to make sure SC locus was located between the two markers.

The authors present results of using RH to overcome SI in a diverse set of germplasm (Fig 2b and Suppl Table 1). I think this work in not critical to the paper. Furthermore, our experience in using RH to transmit SC is mixed. We rarely if any see a 1:1 segregation for SC/SI. We would not classify RH as a primary clone to overcome SC (SI-breaker).

Response:

We thank Reviewer for this suggestion. In early stage of our breeding, we used wild species *S. chacoense* Chc 525-3 (containing the *Sli* gene) as the SC donor rather than RH. But due to the wild background, it caused many linkage drags, such as high levels of toxic steroidal glycoalkaloids and long stolon, which increased difficulties to develop elite inbred line. When we identified the SC gene in RH and mapped it Ch12, we tried to use RH in our breeding and used molecular markers to select the plants carrying the SC gene. Besides, application of RH could also introduce some other desirable traits, such as excellent tuber shape, long-day-adaptive tuberization. Certainly, RH also contains some deleterious alleles (please see our previous publication, Zhou et al., Nature Genetics, 2020, 52: 1018-1023), but we could purge these alleles by molecular selection or genome sequencing. In conclusion, RH is a good SC donor in our breeding program.

According to Kubo et al (2015. PMID: 27246052) there are 14 pollen-specific SLFs in potato. Why this number disagrees with the authors' results and discussion?

Response:

We agree with reviewer that there are usually 14-18 *SLFs* for a certain potato accession. The thirty-two pollen-specific F-box genes we listed are annotated on the whole Chromosome 1, not only within the *S-locus* region. Thus, the number is larger than 14 pollen-specific SLFs reported by Kubo et al. However, *RHC01H2G1617* used as the negative gene in Y2H analysis are locating within the *S-locus* region, and should be considered as a *SLF*.

The authors provide some evolutionary conjecture in the discussion. This model is weak and does not need to be in the manuscript.

Response:

As appearance of *Sli* represents a new way to confer SC to SI plants, which is different from the gain of function mutations of SLFs or loss of function mutations of S-RNase, we think Fig. 4 is a new complement to the two-step model. We add more information to this Figure as suggested by the Reviewer 1. If reviewer insists to remove this figure, we could move it to Supplementary Figures.

The model is modified to show loss of function mutations of *S-RNases* to confer SC as suggested.

BSA-Seq analysis was based on 80 progeny. The number of progeny used is much larger than other previously published studies. Why so many progeny used?

Response:

Thank you for this question. We repeated three times to accurately measure the SC/SI phenotype of the F₁ individuals. In each time, three biological repetitions were used. Thus, we are very confident about the phenotype data of the F₁ individuals, so 40 SC plants and 40 SI plants were selected for the BSA-Seq analysis, and the greater number of individuals used for BSA analysis, the more accurate position result would be anticipated. The result revealed a specific peak where *Sli* locates as expected.

My suggestion is that the authors resubmit when they can provide seed set data to support the transgenic results and address the issues presented above.

Response:

Thank you for your suggestion. The seed data and more details of the *Sli* transgenic lines have been added to the maintext. According to three Reviewer's suggestions, the manuscript has been carefully revised. We are willing to further modify the manuscript if Reviewer has other comments.

Reference:

1. Fossi, M. *et al.* Regeneration of *Solanum Tuberosum* Plants from Protoplasts Induces Widespread Genome Instability. *Plant Physiol.* 2019, 180 (1), 78–86.

REVIEWERS' COMMENTS

Reviewer #1 (Remarks to the Author):

The revised manuscript addresses nearly all points raised in my first review.

However, re. my comment #4 about why F1 PI 225689 x RH produced some a/a progeny, the authors should include their explanation (i.e. that RH may harbor SLFs that recognizes the S-RNase's in PI 225689) in the text. I'm sure I'm not that only reader who would wonder about this.

Also, the way arrows and barred lines are used in Figure 4 could be improved. Currently the figure uses arrows to indicate SLF/S-RNase interaction and barred lines to indicate no interaction. However, in conventional regulatory networks, arrows are typically used to indicate positive regulation, while barred lines are used to represent negative regulation. When an F-box protein recognizes its target protein, the result is a degradation, i.e. a negative regulation of that protein. I think the figure would be more readily understood by nonspecialists if SLF recognition of an S-RNase is indicated with a barred line. Failure to recognize would simply be no line at all.

Reviewer #3 (Remarks to the Author):

General Comment

According to the reviewer's response document, the authors decided to rename the NSF gene to NSF/Sli to avoid any confusion. However, in the text, they used three different ways to name it; Sli, NSF, and NSF/Sli (i.e., P3L23, P4L36, P5L26, P8L11), and therefore there is no consistency in the paper. I agree with Reviewer #1 that the interchangeable use of NSF and Sli is very confusing. I would advocate for the use of Sli throughout since the authors provide evidence that the NSF gene identified in RH is Sli.

In general, there is confusion when the abbreviation of self-compatibility (SC) is used across the text. Sometimes SC is used to define "self-compatible diploid potatoes" as observed in P3L18.

Finally, the discussion section is too vague again, leaving behind important points such as the 536 bp insertion in the NSF/Sli promoter region. I also do not think that the evolution model for SC/SI is needed for this research paper.

Introduction

P2L16 Should say "Potato is consumed as a staple food."

P2L29 This phrase is insufficient to address the complex way in which inbreeding depression influences self-fertility.

P3L1: This additional S-RNase function has been only described in one species of the Rosaceae family, so there is not any proof that it works in Solanum.

P3L15 Eggers did provide detailed methodologies. It is more appropriate to say that the molecular mechanism of Sli was not described.

P3L18, L32: SC is wrongly used, since SC stands for "self-compatibility," as stated in the previous paragraphs.

P3L19 The normal-ness of RH tubers is subjective

P3L19 This makes it seem as if Clot et. al cloned Sli already. Better to say Sli SC haplotype P3L20-22. Better to say "it is unknown whether RH can transmit heritable SC to SI diploid potatoes".

P3L26 This paper did not discover RH.

P3L32: what is the pedigree/species of the PI?

Results

P4L2 Need to say "40 Sc and 40 SI F1 individuals"

P4L10 Explicitly state that RH is A/a

P4L19 adding in this other population is confusing.
P4L27-29: it is not clear which markers were used for fine mapping. M-1 and M-2?
P4L27 remove 'averting' and instead say 'removing the laborious phenotyping requirement'
P5L3 Is S15-65 SI?
P5L13 remove 'typical'
P5L21 Is E SI? Need to establish that SC in E172 is donated by chc 525-3
P5L27-32. It is still unclear whether 125 F1 populations or 125 individual hybrids were being evaluated.
P5L33-34: It is not clear why 15 diploid lines failed to cross with both SC lines due to different flowering times? Can they store the pollen and pollinate those 15 lines when they produce flowers?
P6L19-32 Why did the authors switch to 'Sli' here? Should read 'NSF' for consistency
P6L20 'With the exception of St...'
P6L27 Why was this SLF gene alone selected?
P7L3 When was it established that SLFs that lack PP2 can only interact with a limited number of non-self S-RNases?
P7L13-14 How do these positions tie into genetic bottlenecks?
P7L14 It would be worth discussing the challenges of introducing SC when other fertility traits are not there or flowering time is not synchronized (as observed in this study when attempting to introduce SC to the 125 SI lines.

Discussion

P7L15-33 Rather than belabor the model, it would be more interesting to focus on S-RNase allelic diversity and the prospect of Sli to interact with various S-RNases from a practical breeding standpoint.

Methods

The BSA method should be before, accordingly with the results section.

P8L3 It is still unclear what E172 is. More detail is needed for clone 'E'.
P8L10: Specify that the PI 225689 is SI.
P8L11 How many F2 individuals?
P8L13 What germplasm was phenotyped with this method?
P8L18: "UI" can be confused with unilateral incompatibility.
P8L18 How is it that all three clones having fewer than 1 berry different than all three clones failing to set fruit? I think the English needs to be clarified.
P8L20 How many pistils per genotype?
P8L32 which version of the DM genome?
P9L4: Again, the authors do not explain the nature and origin of markers M1 and M2.
P9L5 multiple F1 populations?
P9L8-9 I am not familiar with the term 'extreme partial separation' Also need to describe what the B/B haplotype represents
P9L34 What is S15-65?
P10L20 Which diploid lines were used for RNAseq? At what developmental stage was style tissue collected for RNAseq?

Figures

Figure 1d: it seems that the SC allele is derived from the line PI 225689.
Figure 2. c) Is the sample in stages 1-4 all floral organs and the pollen sample is mature pollen?
Figure 4 This model is confusing and does not add to the paper
Supplementary Figure 3. This alignment style is very confusing. Why are all three sequences not shown with conservation *?

REVIEWERS' COMMENTS

Reviewer #1 (Remarks to the Author):

The revised manuscript addresses nearly all points raised in my first review. However, re. my comment #4 about why F1 PI 225689 x RH produced some a/a progeny, the authors should include their explanation (i.e. that RH may harbor SLFs that recognizes the S-RNase's in PI 225689) in the text. I'm sure I'm not that only reader who would wonder about this.

Response:

OK. We thank Reviewer for this suggestion, and one sentence was added as suggested to clarify this confusion (Page 4, Line 23-24).

Also, the way arrows and barred lines are used in Figure 4 could be improved. Currently the figure uses arrows to indicate SLF/S-RNase interaction and barred lines to indicate no interaction. However, in conventional regulatory networks, arrows are typically used to indicate positive regulation, while barred lines are used to represent negative regulation. When an F-box protein recognizes its target protein, the result is a degradation, i.e. a negative regulation of that protein. I think the figure would be more readily understood by nonspecialists if SLF recognition of an S-RNase is indicated with a barred line. Failure to recognize would simply be no line at all.

Response:

We thank Reviewer for this suggestion. Figure 4 was modified as suggested.

Reviewer #3 (Remarks to the Author):

General Comment

According to the reviewer's response document, the authors decided to rename the NSF gene to NSF/Sli to avoid any confusion. However, in the text, they used three different ways to name; Sli, NSF, and NSF/Sli (i.e., P3L23, P4L36, P5L26, P8L11), and therefore there is no consistency in the paper. I agree with Reviewer #1 that the interchangeable use of NSF and Sli is very confusing. I would advocate for the use of Sli throughout since the authors provide evidence that the NSF gene identified in RH is Sli.

Response:

We thank Reviewer for this suggestion. As we mentioned in the response to REVIEWER#1, we do agree that 'NSF' creates confusion. NSF was used instead of using of Sli mainly based on two reasons. First, a group from Wageningen used the wild diploid potato species, *Solanum chacoense* Bitt., to clone *Sli* gene, while we used RH as a starting material. Thus, *NSF* was used to differentiate two studies, indicating two group applied different strategies or routes to clone *Sli*. Another reason is we didn't know whether the SC gene (*NSF*) from RH is the same to *Sli* from *Solanum chacoense* Bitt., only after we confirmed that the coding sequences of *NSF* and *Sli* are exactly the same. Once two genes were confirmed to be the same gene, we use *Sli*, not *NSF/Sli*, in the manuscript. We modified the wording to avoid this confusion.

In general, there is confusion when the abbreviation of self-compatibility (SC) is used across the text. Sometimes SC is used to define "self-compatible diploid potatoes" as observed in P3L18.

Response:

We thank Reviewer for this suggestion. We modified the wording as suggested in the revised manuscript.

Finally, the discussion section is too vague again, leaving behind important points such as the 536 bp insertion in the NSF/Sli promoter region. I also do not think that the evolution model for SC/SI is needed for this research paper.

Response:

We thank Reviewer for this suggestion. We modified the discussion section according to the suggestions mentioned here and below (Page 7, Line 5).

Introduction

P2L16 Should say "Potato is consumed as a staple food."

Response:

OK (Page 2, Line 17).

P2L29 This phrase is insufficient to address the complex way in which inbreeding depression influences self-fertility.

Response:

OK, we modified the wording as suggested (Page 2, Line 30-31).

P3L1: This additional S-RNase function has been only described in one species of the Rosaceae family, so there is not any proof that it works in Solanum.

Response:

We thank Reviewer for this suggestion., An example reported in *Nicotiana glauca* was added to the revised manuscript as suggested. (Page 3, Line 2).

P3L15 Eggers did provide detailed methodologies. It is more appropriate to say that the molecular mechanism of Sli was not described.

Response:

OK (Page 3, Line 16).

P3L18, L32: SC is wrongly used, since SC stands for “self-compatibility,” as stated in the previous paragraphs.

Response:

We thank Reviewer for this suggestion. We modified the wording as suggested in the revised manuscript.

P3L19 The normal-ness of RH tubers is subjective

Response:

OK, we deleted ‘normal’ (Page 3, Line 20).

P3L19 This makes it seem as if Clot et. al cloned Sli already. Better to say Sli SC haplotype

Response:

OK (Page 3, Line 21).

P3L20-22. Better to say “it is unknown whether RH can transmit heritable SC to SI diploid potatoes”.

Response:

OK, we thank Reviewer for this suggestion (Page 3, Line 22-23).

P3L26 This paper did not discover RH.

Response:

OK (Page 3, Line 27).

P3L32: what is the pedigree/species of the PI?

Response:

PI 225689 was imported from US GeneBank. Sorry we didn't know its pedigree/species.

Results

P4L2 Need to say “40 Sc and 40 SI F1 individuals”

Response:

OK (Page 4, Line 5).

P4L10 Explicitly state that RH is A/a

Response:

OK, we added this info as suggested (Page 4, Line 4). We also state that PI 225689 is a/a. (Page 3, Line 35)

P4L19 adding in this other population is confusing.

Response:

OK, we deleted the info of the other population.

P4L27-29: it is not clear which markers were used for fine mapping. M-1 and M-2?

Response:

The Indel makers used for fine mapping was provided in the Table S2.

P4L27 remove 'averting' and instead say 'removing the laborious phenotyping requirement'

Response:

OK (Page 4, Line 32).

P5L3 Is S15-65 SI?

Response:

Yes, and this info is added to the revised manuscript (Page 5, Line 8).

P5L13 remove 'typical'

Response:

OK.

P5L21 Is E SI? Need to establish that SC in E172 is donated by chc 525-3

Response:

Yes, and this info is added to the revised manuscript (Page 5, Line 27). We thank Reviewer for this suggestion.

P5L27-32. It is still unclear whether 125 F1 populations or 125 individual hybrids were being evaluated.

Response:

We did 125 hybrids using RH/E172's pollen, and successfully got 110 F₁ populations to evaluate their SC/SI phenotype. We modified the wording to avoid confusion (Page 6, Line 1).

P5L33-34: It is not clear why 15 diploid lines failed to cross with both SC lines due to different flowering times? Can they store the pollen and pollinate those 15 lines when they produce flowers?

Response:

We grew RH/E172 together with 125 self-incompatible lines in field. During flowering time, pollens of RH or E172 were collected and used to pollinate the 125 lines. As the flowering time of 15 lines were obviously different from other lines, or failed to flower or have few flowers, we did store the pollen and pollinate on those 15 lines but didn't get fruits. Thus, we speculated that the failed phenotype change from SI to SC for the 15 lines may be caused by unsuccessful pollination.

The possibility that whether *Sli* can not recognize the *S*-RNase harbored by these lines needs to be further explored in future.

P6L19-32 Why did the authors switch to ‘*Sli*’ here? Should read ‘NSF’ for consistency

Response:

As mentioned earlier, once we proved that two genes (*NSF/Sli*) are the same gene, we use *Sli*, not *NSF/Sli*, in the manuscript.

P6L20 ‘With the exception of St...’

Response:

Sorry to make this mistake (Page 6, Line 27).

P6L27 Why was this *SLF* gene alone selected?

Response:

This *SLF* gene is closed to the *S*-RNase and specific expressed in pollen. This info was provided in the legend of Supplementary Fig. 6.

P7L3 When was it established that *SLFs* that lack *PP2* can only interact with a limited number of non-self *S*-RNases?

Response:

OK, we thank Reviewer for this suggestion. We here just want to emphasize the importance of *PP2* domain for the interactions of *Sli* and *S*-RNases. We deleted this sentence to avoid confusion.

P7L13-14 How do these positions tie into genetic bottlenecks?

Response:

To create self-compatible lines, either loss of function mutations of the *S*-RNase or introduction of *Sli* is required. The genetic bottlenecks may be caused by the deleterious mutations linked to these loci. Since *S*-RNase is near centromere that has a lower recombination rate than that of telomere, it would be difficult to get rid of these deleterious mutations. Thus, we think usage of *SC* genes from different sources in breeding can effectively avoid the unexpected genetic bottlenecks. We modified the wording to avoid this confusion (Page 7, Line31).

P7L14 It would be worth discussing the challenges of introducing *SC* when other fertility traits

are not there or flowering time is not synchronized (as observed in this study when attempting to introduce SC to the 125 SI lines.

Response:

OK, we modified the discussion section as suggested.

Discussion

P7L15-33 Rather than belabor the model, it would be more interesting to focus on S-RNase allelic diversity and the prospect of Sli to interact with various S-RNases from a practical breeding standpoint.

Response:

OK, we modified the discussion section as suggested.

Methods

The BSA method should be before, accordingly with the results section.

Response:

OK.

P8L3 It is still unclear what E172 is. More detail is needed for clone 'E'.

Response:

E was provided by Wageningen University and was described in this paper: A genetic map of potato (*Solanum tuberosum*) integrating molecular markers, including transposons, and classical markers. This info was added to the revised manuscript (Page 8, Line9).

P8L10: Specify that the PI 225689 is SI.

Response:

OK (Page 8, Line14).

P8L11 How many F2 individuals?

Response:

6624 individuals. This info was added to the revised manuscript (Page 8, Line16).

P8L13 What germplasm was phenotyped with this method?

Response:

The F₁ hybrids from RH crossed with PI 225689 (PI 225689×RH) or RH/E172 crossed with the 125 lines were phenotyped. We added this info to the revised manuscript (Page 8, Line18-19).

P8L18: “UP” can be confused with unilateral incompatibility.

Response:

OK. We used ‘U’ instead of ‘UP’ as suggested (Page 8, Line25).

P8L18 How is it that all three clones having fewer than 1 berry different than all three clones failing to set fruit? I think the English needs to be clarified.

Response:

Sorry, we modified the wording in revised manuscript (Page 8, Line24).

P8L20 How many pistils per genotype?

Response:

Five pistils. This info was added to the revised manuscript (Page 9, Line28).

P8L32 which version of the DM genome?

Response:

It is DM V4.03. This info was added to the revised manuscript (Page 8, Line34).

P9L4: Again, the authors do not explain the nature and origin of markers M1 and M2.

Response:

Two Indel markers M1 and M2 were developed based on the re-sequencing data of RH and PI 225689. This info was added to the revised manuscript (Page 9, Line13-14).

P9L5 multiple F₁ populations?

Response:

Sorry, only one F₁ population (created by a cross between PI 225689 and RH) was used in the fine mapping. This info was modified to the revised manuscript (Page 9, Line5).

P9L8-9 I am not familiar with the term ‘extreme partial separation’ Also need to describe what the B/B haplotype represents

Response:

We thank Reviewer for this suggestion, and replaced the ‘extreme partial separation’ with ‘extreme segregation distortion’. B/B haplotype represents a/a, and we modified the wording to avoid confusion (Page 9, Line8-9).

P9L34 What is S15-65?

Response:

S15-65 is a diploid self-incompatible line from *S. tuberosum* group Phureja. We imported this line from CIP. This info was added to the revised manuscript (Page 9, Line21).

P10L20 Which diploid lines were used for RNAseq? At what developmental stage was style tissue collected for RNAseq?

Response:

The mature style collected from the 125 lines were used for RNA-Seq. This info was added to the revised manuscript (Page 10, Line29).

Figures

Figure 1d: it seems that the SC allele is derived from the line PI 225689.

Response:

OK, we modified Fig. 1d as suggested. We thank Reviewer for this suggestion.

Figure 2. c) Is the sample in stages 1-4 all floral organs and the pollen sample is mature pollen?

Response:

Yes.

Figure 4 This model is confusing and does not add to the paper

Response:

We modified Fig. 4 as suggested by Reviewer #1. As Sli is not a tradition *S-locus* F-box protein and works like a general S-RNase inhibitor to confer SC, we think using a model to show the

molecular mechanism of Sli is more straightforward and would highlight the major discovery of this story.

Supplementary Figure 3. This alignment style is very confusing. Why are all three sequences not shown with conservation *?

Response:

OK, we modified this Figure as suggested.